# Single-cell microRNA-mRNA co-sequencing reveals non-genetic heterogeneity and mechanisms of microRNA regulation

Nayi Wang[1,2,3], Ji Zheng[2,3,4], Zhuo Chen[1,2,3], Yang Liu[2,3], Burak Dura[1], Minsuk Kwak[1], Juliana Xavier-Ferrucio[3,5], Yi-Chien Lu[3,5], Miaomiao Zhang[2,4,6], Christine Roden[2,3], Jijun Cheng[2,3], Diane S. Krause[3,5], Ye Ding[7], Rong Fan [1,3] & Jun Lu[2,3,8,9]

Measuring multiple omics profiles from the same single cell opens up the opportunity to decode molecular regulation that underlies intercellular heterogeneity in development and disease. Here, we present co-sequencing of microRNAs and mRNAs in the same single cell using a half-cell genomics approach. This method demonstrates good robustness (~95% success rate) and reproducibility ($R^2 = 0.93$ for both microRNAs and mRNAs), yielding paired half-cell microRNA and mRNA profiles, which we can independently validate. By linking the level of microRNAs to the expression of predicted target mRNAs across 19 single cells that are phenotypically identical, we observe that the predicted targets are significantly anti-correlated with the variation of abundantly expressed microRNAs. This suggests that microRNA expression variability alone may lead to non-genetic cell-to-cell heterogeneity. Genome-scale analysis of paired microRNA-mRNA co-profiles further allows us to derive and validate regulatory relationships of cellular pathways controlling microRNA expression and intercellular variability.

[1] Department of Biomedical Engineering, Yale University, New Haven, CT 06520, USA. [2] Department of Genetics, Yale University School of Medicine, New Haven, CT 06510, USA. [3] Yale Stem Cell Center, Yale Cancer Center, New Haven, CT 06520, USA. [4] Department of Urology, Southwest Hospital, Third Military Medical University, 400038 Chongqing, China. [5] Department of Laboratory Medicine, Yale University School of Medicine, New Haven, CT 06510, USA. [6] School of Medicine, Jiangsu University, 212013 Zhenjiang, Jiangsu, China. [7] Wadworth Center, New York State Department of Health, Albany, NY 12208, USA. [8] Yale Center for RNA Science and Medicine, New Haven, CT 06520, USA. [9] Yale Cooperative Center of Excellence in Hematology, New Haven, CT 06520, USA. Correspondence and requests for materials should be addressed to R.F. (email: rong.fan@yale.edu) or to J.L. (email: jun.lu@yale.edu)

Small RNAs have emerged as important non-coding regulators in diverse biological settings. The most widely studied small RNAs are miRNA that regulate protein-coding expression post-transcriptionally[1–3]. MiRNA expression profiles in tissues or cell populations are highly informative to reveal cellular states, such as in human cancers[2,4,5] and identify cellular mechanisms[6,7]. The advances of single-cell genomics have raised the prospect that single-cell miRNA profiles could add another dimension to small RNA research by facilitating the molecular understanding of intercellular heterogeneity. Unlike single-cell RNAseq techniques, which have matured considerably in the past years[8–17], single-cell small RNA sequencing has been demonstrated very recently and applied to differentiating naive versus primed human embryonic stem cells and examining intercellular heterogeneity of miRNA expression[18], with the latter confirming previous findings of single-cell miRNA measurements by lower throughput methods such as quantitative RT-PCR or fluorescence in situ hybridization (FISH)[19–21]. However, given the roles of small RNAs to modulate gene expression post-transcriptionally, for example, via miRNA-mediated degradation of target transcripts, it is important to measure both small RNAs and mRNAs at the genome-scale in order to decipher the mechanism underlying observed intercellular miRNA and mRNA heterogeneity. We thus envision that obtaining miRNA and mRNA profiles from the same single cells could empower studying how miRNAs modulate non-genetic cell-to-cell variability post-transcriptionally and how miRNA variability could be modulated via protein-coding genes.

Over the past years, technology breakthroughs in single-cell omics have enabled genome-wide profiling of genetic mutations[22], copy number variation[23,24], DNA methylation[24,25], chromatin-accessibility[26], and gene expression at the transcriptional level[27]. Combining two or three of these measurements on the same single cells has been demonstrated[24,28–34], but almost exclusively on co-analysis of genomic DNA and mRNAs to correlate pre-transcriptional alterations to mRNA gene expression. Genome-wide profiling of both mRNAs and post-transcriptional regulators such as miRNAs remains challenging, due in part to the incompatibility of current mRNA-seq and small RNA sequencing protocols[35]. Strategies such as separation of miRNAs from mRNAs are not yet reliable for single-cell co-measurement of both types of RNAs at the genome-scale.

We report here the co-sequencing of single-cell small RNA and mRNA transcriptomes using a half-cell genomics approach, which has been built upon our prior success in direct capture and profiling of miRNAs in low-quantity whole cell lysate[2] and the rigorous validation of half-cell sampling to reliably represent the whole cell RNA expression levels. While previous efforts have examined the relationships between miRNA and mRNA expression profiles in bulk samples[36–40], we demonstrate that decoding single-cell-level small RNA and mRNA profiles is feasible to reveal regulatory mechanisms both upstream and downstream of intercellular miRNA heterogeneity. This work demonstrates single-cell miRNA and mRNA co-sequencing at the genome-scale and opens up opportunities to investigate how the changes in small RNA expression in single cells contribute to non-genetic cell-to-cell variability and how to perturb such relationships for controlling cellular function in a heterogeneous population.

## Results

**Strategy for co-profiling of single-cell miRNAs and mRNAs.** To achieve the goal of profiling small RNAs and mRNAs from the same single cells, we utilized a half-cell genomics approach in which a single cell was lysed and the lysate was split evenly into two half-cell fractions, with each fraction subjected to either miRNA or mRNA transcriptome sequencing (Fig. 1a–c). For profiling miRNA expression in half cells, we modified our previous method[2] to develop a protocol for generating small RNA

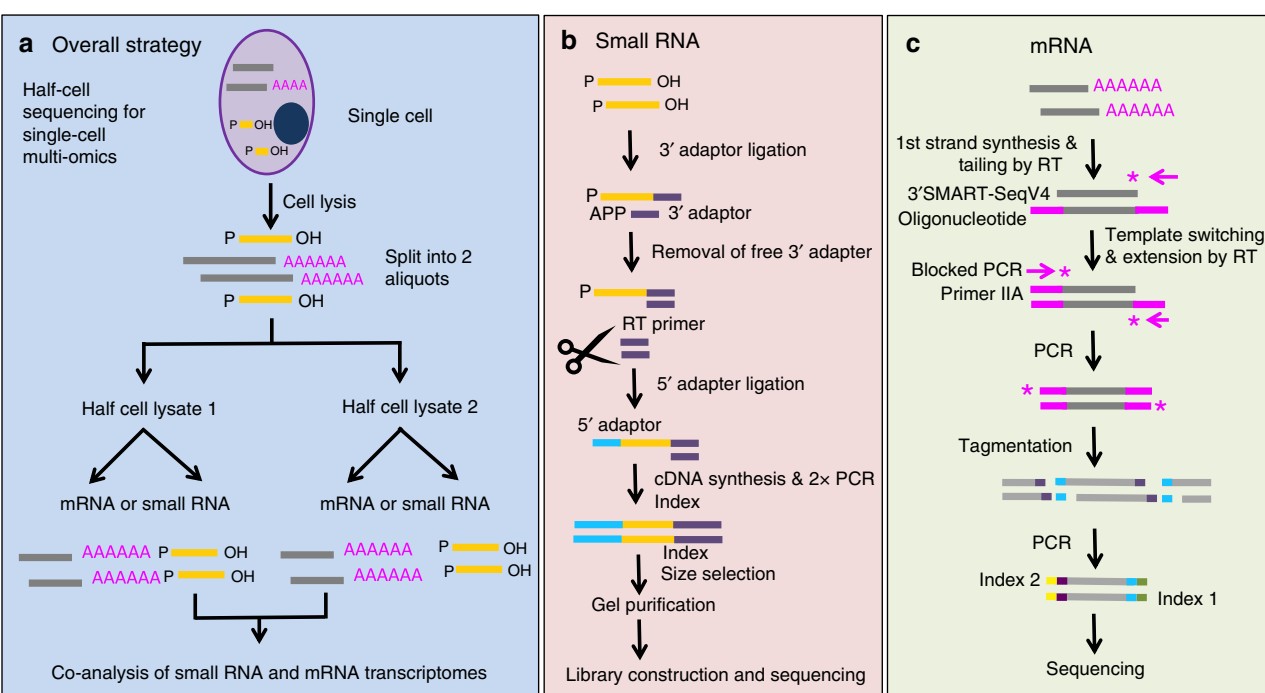

**Fig. 1** Experimental workflow. **a** Overall strategy for profiling miRNA and mRNA from the same single cells using half-cell genomics. It involves cell lysis, half-cell split, followed by small RNA & large RNA library preparation. **b** For miRNA library preparation, a pre-adenylated (APP) 3′ adaptor was used to ligate to the 3′ end of miRNA molecules, followed by digestion of unreacted 3′ adaptor, ligation with 5′ adaptor, RT and PCR amplification. **c** For mRNA library preparation, first-strand cDNA synthesis was primed by the 3′ SMART-Seq CDS Primer IIA. Template switching at the 5′ end of transcript was performed using the SMART-Seq v4 oligonucleotides. After PCR amplification, cDNA was fragmented using Illumina's Tagmentation process

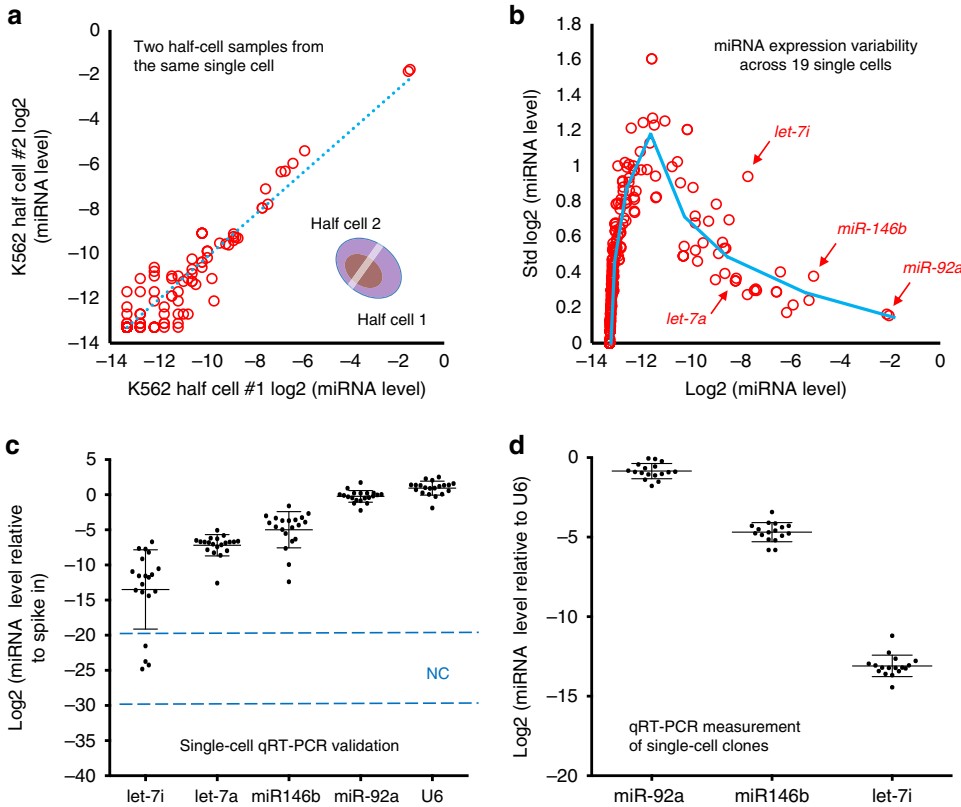

**Fig. 2** Profiling of miRNAs from half-cell lysate. **a** A single K562 cell was lysed, and two halves of the lysate were separately subjected to small RNA sequencing. Scatter plot of normalized and log2-transformed miRNA expression levels (miRNA expression levels represent the fraction among total miRNA, see Methods) is shown. Each circle represents one annotated miRNA. The correlation coefficient $R^2 = 0.93$. **b** The standard deviations (on y-axis) of miRNA expression across 19 successfully profiled half cells were plotted against the mean expression on x-axis, using normalized and log2-transformed miRNA expression data as described in **a**. Each circle represents one annotated miRNA. The blue line shows the trend of the distribution based on locally averaged values. Specific miRNAs are highlighted with red arrows. **c** Validation experiment was carried out by quantifying miRNA expression in 20 single K562 cells using qRT-PCR. Relative levels of let-7i-5p, let-7a-5p, miR-146b-5p, miR-92a-3p, and U6 were determined relative to a spike-in control (synthetic miR-371). Each dot represents one K562 cell. Each data point reflects the average measurements from two technical replicates. Error bars present standard deviation. $N = 20$. The levels of miRNA are shown in log2 scale. The dashed lines and NC (negative control) indicates the detection range of water samples. **d** Single K562 cells were grown to derive 17 clones, and the relative levels of miR-92a-3p, miR-146b-5p, and let-7i-5p were determined in clonal populations by qRT-PCR. Roughly 40,000 cells were used from each clone. Each dot represents one K562 clone. Each data point reflects the average from two technical replicates. Error bars present standard deviation. $N = 17$. U6 was measured as a control for normalization. The levels of miRNAs are shown in log scale

sequencing library from single cells or half-cell-level materials. In this protocol, single-cell lysate was heated to release miRNAs for sequential ligation reactions with 3′- and 5′-molecular adaptors and then the ligated products were reverse transcribed, and amplified with polymerase chain reaction (PCR) for library preparation and next-generation sequencing (Fig. 1b). For profiling mRNAs, we used SMART-Seq (v4), a commercialized kit for manual processing of low-input or single-cell poly-adenylated mRNAs to generate full-length libraries for RNA sequencing (Fig. 1c). As detailed below, we successfully developed, combined, and validated these methods to generate sequencing libraries from half-cell materials and the half-cell miRNA and mRNA profiles faithfully capture single-cell transcriptomes.

**Evaluating the profiling of half-cell miRNA expression.** Although single-cell omics technologies have been developed to quantify intercellular heterogeneity via comparing measured molecular profiles between individual cells and the bulk data, it remains challenging to determine how much variability is true biological heterogeneity and how much is attributable to technical noise associated with the processing of single cells. The best

experiment to address this question is to measure variability between two half-cell materials split evenly from the same single cell. We first evaluated this approach to profiling miRNAs. A single K562 human acute myeloid leukemia cell was transferred to and lysed in a PCR tube. The lysate (total volume ~10 μl) was eventually split into two halves into two PCR tubes to independently perform small RNA capture, amplification, and preparation of sequencing libraries as described in Fig. 1b. The sequencing data from each of these two halves were compared against each other for miRNA expression, which gave rise to an $R^2$ value of 0.930 (Fig. 2a), indicating low technical noise and high consistency.

Of note, initial tests with even splitting (1:1 in vol) of single-cell lysate prepared with standard cell lysis protocol did not yield even splitting of miRNAs in two half-cell samples. Interestingly, we observed selective enrichment or depletion of a group of miRNAs in one half-cell sample presumably due to the binding to intracellular proteins. This remains to be further investigated. In this work, we modified the lysis protocol in several steps (see Methods) including introducing freeze/thaw and heat treatment, which eventually led to a robust procedure to evenly split

miRNAs in two half-cell samples (Supplementary Figure 1). We further examined the validity of this half-cell approach using different cell types and another measurement technique. Small RNA libraries were generated from two halves of a single murine hematopoietic BaF3 cell, or single K562 and 293T cells, and measured with a Luminex bead array assay. The results again exhibited a high degree of concordance between these two half cells (Supplementary Figure 2a). While two half-cell samples from the same single cell gave rise to high concordance in miRNA profiles, different single cells obtained from the same cell population exhibited a higher degree of variability by both qRT-PCR analysis and expression profiling (Supplementary Figures 3, 4),

Upon validating the consistency of the half-cell approach to capture single-cell miRNA profiles, we turned to measure intercellular variability by sequencing 20 half-cell K562 miRNA samples. The other halves of the same cells were processed for RNAseq (to be described in next section). The sequencing data from 19 single cells passed the quality check for both miRNAs and mRNAs. The one that failed the quality assessment had very low levels of mappable reads in both miRNA and mRNA data presumably due to the poor quality of this cell itself (e.g., cell damaged in the capture and transfer process). Principal component analysis (PCA) of these 19 samples and 2 half-cell samples derived from the same cell further confirmed high concordance in two half-cell splits and the variability among single cells is higher, suggesting the existence of intercellular heterogeneity of miRNA expression (Supplementary Figure 5). Overall, the success rate (95%) to obtain quality sequencing libraries for both miRNA and mRNA using the half-cell genomics approach was excellent.

The intercellular variability of miRNA expression could be directly observed in our data either by a correlation matrix (Supplementary Figure 6) or by quantifying the standard deviation of log2-transformed miRNA expression versus the mean expression across 19 half cells (Fig. 2b). As expected, low expression level miRNAs showed inherently large standard deviations in log2 data. The variation of high abundance miRNAs gradually decreased as the expression level increased. MiR-92a-3p was detected as the most abundant miRNA in our K562 cell sample, in agreement with previous literature[40]. Comparing intercellular variability across miRNAs, both miR-146b-5p and let-7i-5p showed higher variability than miRNAs with comparable mean expression levels (for miR-146b-5p, std = 0.377; for let-7i-5p, std = 0.939). For example, let-7a-5p, a miRNA in the same family as let-7i-5p, had comparable mean expression to let-7i-5p, but with much lower variability (for let-7a-5p, std = 0.351). These results showed the direct evidence of intercellular miRNA expression heterogeneity, which differed substantially for individual miRNAs.

To validate the observed intercellular heterogeneity, we performed single-cell qRT-PCR to measure four selected miRNAs, let-7a-5p, let-7i-5p, miR-146b-5p, and miR-92a-3p, in a separate batch of cells (Fig. 2c). Indeed, the variations from single-cell qRT-PCR paralleled those observed in half-cell small RNA sequencing data: miR-92a-3p and let-7a-5p show relatively lower levels of variation (standard deviation std = 0.83 and std = 1.51, respectively), miR-146b-5p an intermediate level of variation (std = 2.57), and let-7i-5p a high level of variation (std = 5.64) (Fig. 2c). We further determined that these variations could not be fully attributed to genetic differences but instead associated with non-genetic cellular heterogeneity. We generated 17 K562 single-cell clones and measured the same miRNAs using qRT-PCR for each clone (~40,000 cells per clone). One would expect that if the miRNA expression heterogeneity in single cells was

driven by genetic differences between cells, large variation would be manifested similarly between single-cell clonal populations because each clone was derived from a single cell and cells within a clone share the genetic variations present in the initial single cell. In contrast, we observed low levels of variability cross these single-cell clones (for miR-92a-3p, std = 0.48; for let-7i-5p, std = 0.68; for miR-146b-5p, std = 0.60) (Fig. 2d). Collectively, these data support that the half-cell small RNA profiles could faithfully capture single-cell miRNA variability, which allowed us to use half-cell small RNA sequencing to interrogate non-genetic intercellular miRNA heterogeneity.

**Evaluating the profiling of half-cell mRNA expression.** As described above, the 20 single K562 cells processed for half-cell small RNA sequencing were also analyzed for mRNA transcriptomes using the remaining half-cell lysates. SMART-seq was reported to be of high coverage, low bias, and good reproducibility compared to other methods when single-cell data were compared to population, spike-in controls or theoretical predictions[41]. However, the true technical variability cannot be accessed unless the same cell can be analyzed multiple times for RNAseq, which has not been practical. Herein, we conducted SMART-seq with two halves of a single K562 cell and the gene expression profiles were compared (Fig. 3a), which yielded a Pearson correlation $R^2$ value of 0.930, which provided a direct evidence about the level of technical noise. Similar results were observed with human 293T and murine NIH3T3 cells (Supplementary Figure 2b). These data supported the validity to use half-cell sequencing data to represent single-cell mRNA transcriptome.

We next examined the variation of mRNA expression across 19 half K562 cells (Supplementary Figure 7). The plot (Fig. 3b) showing mRNA variation across 19 half cells as a function of the mean gene expression levels followed the anticipated trend that the log2 variation became lower as the levels of gene expression increased. Unsupervised consensus clustering was conducted to detect transcriptional states with similar profiles and resulted in three major clusters (Fig. 3c). The first cluster of cells was enriched for genes associated with translational processes but depleted for genes associated with processes such as acetylation. The second cluster of cells were opposite, depleted for translation-related genes but enriched for the acetylation process (Supplementary Data 1). The third cluster of cells contain both signatures of the first and second clusters and are likely more differentiated cells with the increased expression of erythroid lineage genes.

In order to further validate the observed clusters, we used a high-throughput single-cell 3′-end mRNA-seq technology[42–44] to generate the mRNA transcriptome data from hundreds of single K562 cells. Quality check was performed and compared against published single-cell RNAseq data for the same cell line (Supplementary Figures 8 and 9). Unsupervised consensus clustering analysis again yielded three major clusters with the proportions (40.55: 42.83: 16.62%) similar as the 19 single-cell data (36.84: 47.37: 15.79%) (Fig. 3c, d). The analysis using t-distributed stochastic neighbor embedding (t-SNE) was performed to visualize the major clusters and the gene expression distribution (Supplementary Figure 10). Pathway analysis further confirmed that these clusters were enriched for the same biological processes, notably the translational activities and the acetylation pathway (Fig. 3c, d and Supplementary Data 1 & 2). Therefore, the mRNA transcriptional phenotypes detected by the half-cell genomics approach and single-cell 3′ mRNA-seq were in good agreement, justifying the validity to use half-cell mRNA profiles to capture single-cell transcriptome and to examine single-cell gene expression heterogeneity.

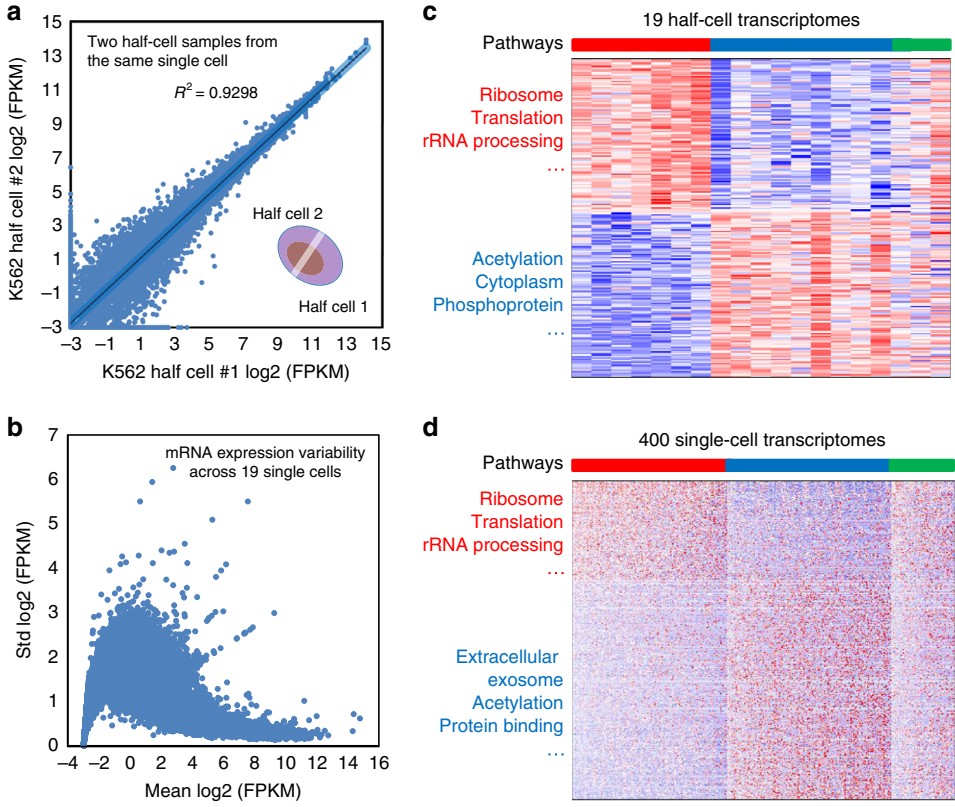

**Fig. 3** Global profiling of mRNAs from half cells. **a** A single K562 cell was lysed, and two halves of the lysate were separately subjected to RNAseq. Scatter plot of normalized and log2-transformed mRNA expression levels (in FPKM units) is shown. Each circle represents one annotated mRNA. The correlation coefficient $R^2$ is 0.930. **b** The standard deviations of mRNA expression across 19 successfully profiled half cells were plotted against the mean expression, using log2-transformed mRNA expression data. Each circle represents one annotated mRNA. **c** Consensus clusters were identified in the 19 K562 half-cell mRNA expression data. A heatmap is shown for differentially expressed genes between the red and blue clusters, with enriched pathways annotated by the DAVID pathway analysis. Each row represents one annotated gene whereas each column represents a single cell. **d** K562 cells were profiled using a massively parallel single-cell 3′-end RNAseq technology. A heatmap is shown for differentially expressed genes between the red and blue consensus clusters, with enriched pathways annotated by the DAVID pathway analysis. Each row represents one annotated gene whereas each column represents a single cell. Within the heatmaps, blue color indicates low expression whereas red indicates higher expression

**miRNA and miRNA-target relationship on the single-cell level**. Having both miRNA and mRNA profiles obtained from the same single cells, we reasoned that if miRNA expression heterogeneity contributed to the control of single-cell level mRNA expression heterogeneity, one would expect that the mRNA targets to be anti-correlated with miRNA expression levels. It has been known that the level of global miRNA expression is low in cancer cells[4], particularly cancer cell lines such as K562, we thus focussed first on the most abundantly detected miRNA miR-92a-3p. We ranked all mRNAs based on their Pearson correlation to the miRNA, and asked whether predicted targets of miR-92a-3p were more anti-correlated to the miRNA than non-targets. Using two independently defined miRNA-target predictions (TargetScan and MSigDB), we observed that both target sets were significantly enriched toward the negative correlation scores (Fig. 4a). In contrast, predicted targets of another miRNA, miR-125a-5p, were not enriched for correlation with miR-92a-3p (Fig. 4a). These data support that the single-cell-level expression of miRNA targets was overall more negatively correlated to miRNA expression, and suggest that miRNA expression variation contributes to the control of mRNA variability.

While examining the predicted targets of miR-92a-3p, we noticed an interesting relationship between the level of target expression and the likelihood of the target being negatively correlated with the miRNA. Specifically, for targets of higher abundance (mean log2(RPKM) > 4), we observed a much

stronger and significant enrichment toward negative correlation than targets with lower expression (Fig. 4b). This relationship could also be observed when analyzing correlation between miR-125a-5p and its predicted targets, as well as between miR-26a-5p and its targets (Fig. 4c, d). In both cases, only targets with mean log2(RPKM) over 4 showed significant enrichment toward negative correlation with the corresponding miRNA, whereas targets with lower expression were not. Indeed, if all the targets of miR-125a-5p and miR-26a-5p were analyzed without categorization based on their expression levels, we could not detect any significant enrichment. This was obviously due to predicted targets of lower expression diluting out the signal for abundantly expressed targets, the latter of which were of a small fraction among all predicted targets. See Discussion for the possible explanations of this observed correlation. Taken together, the data above suggest that miRNA heterogeneity contributes to shaping the mRNA variability in single cells, particularly for abundantly expressed targets.

The three miRNAs mentioned above, including miR-92a-3p, miR-125a-5p, and miR-26a-5p, shared the common properties that they were of relatively high abundance in K562 cells and they had a reasonably long list of predicted targets (>500) to allow effective stratification by expression levels. We thus examined miRNAs with lower abundance to determine if targeting relationships could also be observed in paired half-cell miRNA and mRNA profiles. For let-7a-5p, which was weakly expressed in

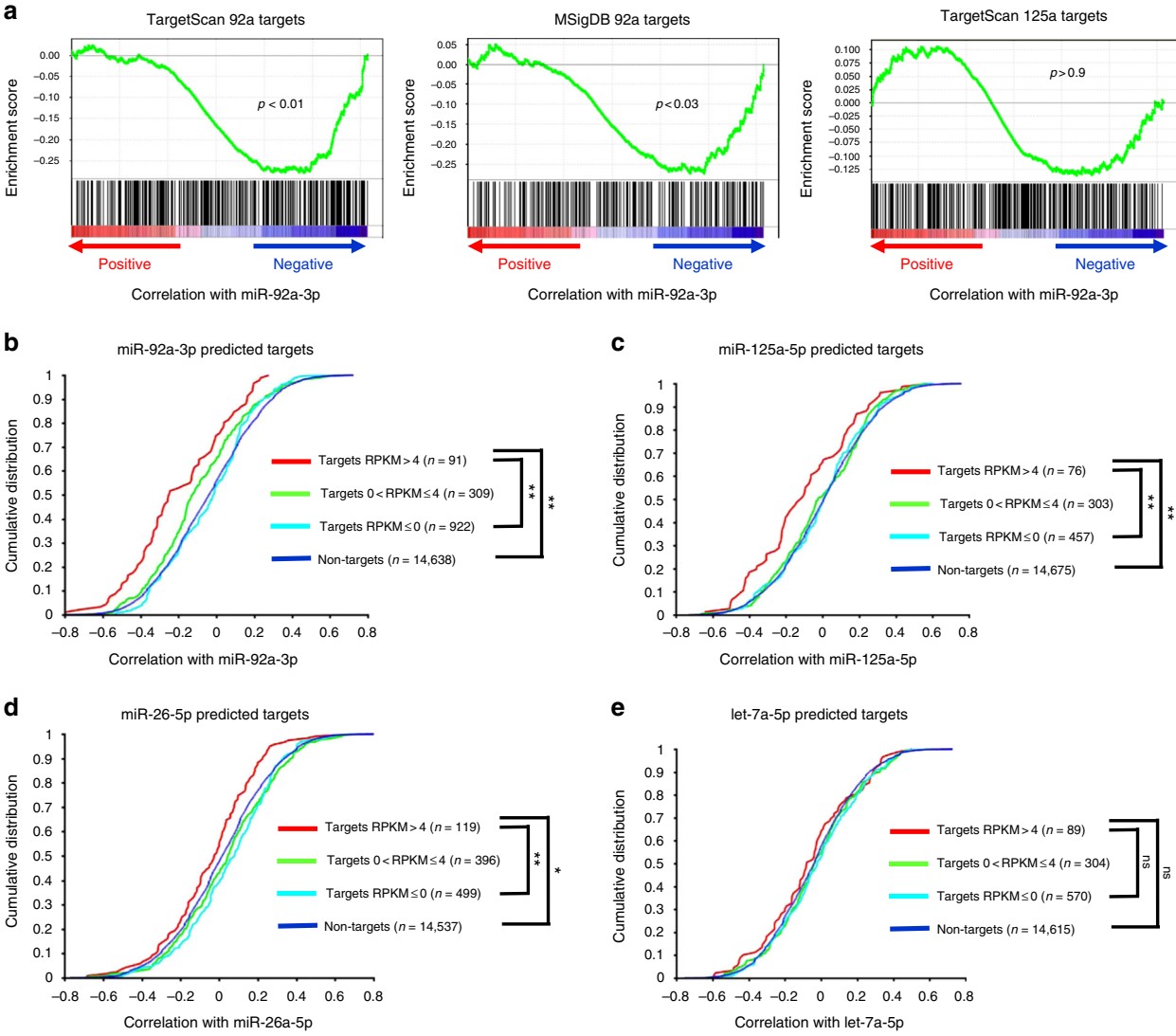

**Fig. 4** Relationships between miRNA and miRNA targets in paired half-cell data. **a** mRNAs were ranked according to their correlation with miR-92a-3p using data from the 19 pairs of half-cell miRNA and mRNA profiles. The resultant rank list was queried with predicted targets of miR-92a-3p obtained from TargetScan or MSigDB (left and middle panels) or with targets of miR-125a-5p from TargetScan (right panel). Gene set enrichment analysis plots are shown with P-values indicated. **b–e** mRNAs were ranked according to their correlation with **b** miR-92a-3p, **c** miR-125a-5p, **d** miR-26a-5p, or **e** let-7a-5p. Cumulative distribution functions were plotted with non-targets and with predicted targets of the corresponding miRNA (based on TargetScan). Predicted targets were further categorized based on the mean RPKM values across the 19 K562 half-cell samples. The number of genes in each category is indicated in parentheses. P-values were calculated based on the Kolmogorov–Smirnov test between the indicated pairs of conditions. **\*\*P < 0.005; \*P < 0.05; ns not significant

K562 cells (Fig. 2b, c), we did not observe significant enrichment of targets toward negative correlation with the miRNA, even though a trend could be seen for predicted targets with abundant expression levels (Fig. 4e). This was not surprising, as it has been reported that functional targeting requires at least several hundred miRNA molecules per cell[45]. Likewise, we did not observe significant targeting relationships for let-7i-5p and miR-146b-5p, presumably due to low abundance (for let-7i-5p) or a short list of predicted targets (for miR-146b-5p).

Taken together, the data above demonstrate the feasibility of examining functional miRNA targeting directly in paired half-cell miRNA and mRNA profiles, with the targeting relationship on the single-cell level being dependent on both the level of miRNA expression and the level of its targets. The data suggest the possibility for abundantly expressed miRNAs to influence global transcriptomic phenotypes in single cells.

**Regulation of miRNA expression heterogeneity**. It is conceivable that heterogeneity in cells' molecular states may contribute to miRNA expression heterogeneity. We thus attempted the search for regulatory relationships that control miRNA gene expression utilizing the cell-to-cell variability of miRNA and mRNA in single cells. We specifically focused on the two highly variable miRNAs, miR-146b-5p and let-7i-5p, and queried the mRNA genes (e.g., using gene set enrichment analysis (GSEA) based upon the MSigDB database[46]) or molecular connectivity (e. g, using the Connectivity Map (CMap)[47]) in order to reveal cellular pathways and chemical perturbations that are capable of modulating the expression of the corresponding miRNA (Fig. 5a).

For miR-146b-5p, we ranked mRNA genes according to their correlation with the miRNA, and performed GSEA querying MSigDB[46,48]. The most noticeable association was a negative relationship between multiple gene sets for protein translation

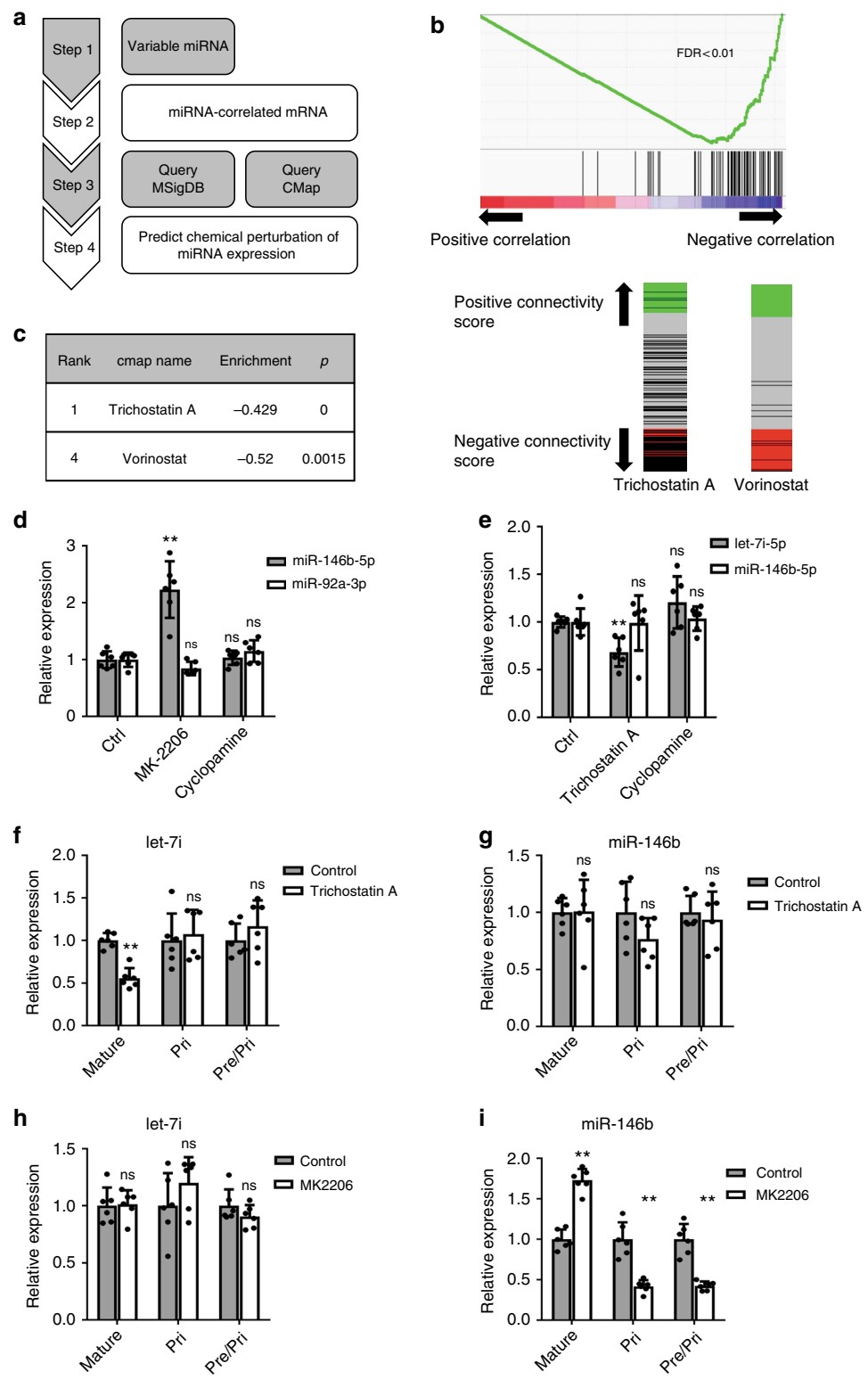

and miR-146b-5p (Fig. 5b and Supplementary Data 3). Interestingly, protein translation was also identified as a molecular process differentially expressed between the major mRNA transcriptomic subtypes (Fig. 3c). We hypothesized that a common upstream regulator controls cellular translation and miR-146b-5p expression in opposite ways. Given that the AKT pathway positively regulates cellular translation, we thus tested the hypothesis that inhibition of AKT could elevate miR-146b-5p

expression. We treated K562 cells with an allosteric AKT inhibitor MK-2206, which significantly increased miR-146b-5p expression (Fig. 5d). A similar increase was observed in MCF-7 cells (Supplementary Figure 11a). In contrast, MK-2206 did not affect miR-92a-3p or let-7i-5p (Fig. 5d, h), and the inhibition of the hedgehog pathway by cyclopamine or the inhibition of HDAC by Trichostatin A did not significantly affect miR-146b-5p expression in K562 cells (Fig. 5d, g), indicating the specificity at

**Fig. 5** Uncovering regulators of miRNA expression through paired half-cell data. **a** Schematics of the approach to infer chemical perturbation of miRNA expression through paired half-cell mRNA and miRNA profiles. Regulators of miRNA were predicted based on gene set enrichment analysis (GSEA) Molecular Signature Database (MSigDB) and Connectivity Map (CMap). **b** mRNAs were ranked according to Pearson correlation coefficients with miR-146b-5p. The resultant rank list was used to query the MSigDB database. An enrichment plot is shown for the gene set KEGG-Ribosome in which genes in this gene set were enriched for negative correlation with the miRNA. False discovery rate (FDR) is indicated. **c** mRNAs were ranked according to Pearson correlation coefficients with let-7i-5p. mRNAs with correlation coefficient $R > 0.45$ were defined as the positive signature whereas mRNAs with correlation coefficient $R < -0.45$ were used as the negative signature. mRNA signatures of let-7i-5p were used to query the CMap database. Two HDAC inhibitors were ranked at the top of the list (left panel). Right panel shows the enrichment plots of Trichostatin A and Vorinostat treatment instances within the CMap database. **d–i** K562 cells were treated with 10 µM AKT inhibitor MK-2206, 10 µM Cyclopamine, 1 µM HDAC inhibitor Trichostatin A or vehicle control (DMSO) for 24 h. **d, e** The expression levels of miR-146b-5p, miR-92a-3p, or let-7i-5p were determined with qRT-PCR. $N = 6$ biological replicates. Error bars: standard deviation. A representative experiment out of two is shown. **f–i** Mature miR-146b-5p and let-7i-5p levels, as well as the expression levels of the corresponding primary miRNA transcript were determined by qRT-PCR. In addition, primer sets that detect both primary and precursor miRNAs were also used (pri/pre). $N = 6$ biological replicates. Error bars: standard deviation. A representative experiment out of two is shown. *$P < 0.01$; **$P > 0.05$; ns not significant, as determined by Student's $t$-test

the levels of both miRNA and cellular pathway. Upon AKT inhibition of K562 cells, we noticed that the intercellular variability of miR-146b-5p expression was reduced (Supplementary Figure 11b, for control std = 2.41, for AKT inhibition std = 1.50; $P < 0.006$ by permutation tests, see Methods). We further noticed that the AKT inhibitor elevated the expression of mature miR-146b-5p, but decreased the expression of the primary transcript of miR-146b (Fig. 5i), whereas no change was observed for the pri-let-7i or mature let-7i-5p (Fig. 5h), suggesting that the AKT pathway specifically inhibits the post-transcriptional maturation of miR-146b. Similar results were observed when K562 cells were treated with two additional AKT inhibitors (Supplementary Figure 11c, d). These data indicate that a previously unknown regulatory relationship for miR-146b-5p expression could be uncovered by analyzing the correlation between miRNA and mRNA profiles in single cells, and support a role for AKT to control intercellular variability of miRNA expression.

For mRNAs correlated with let-7i-5p, we did not find statistically significant positive or negative enrichment for any gene set in MSigDB after multiple-hypotheses correction (Supplementary Data 3). Instead, we defined the positive and negative transcriptomic signatures of let-7i-5p by using mRNAs correlated and anti-correlated with this miRNA respectively, and used these signatures to query the CMap, a database of gene expression responses in the presence of chemical perturbation[47]. Two histone-deacetylase (HDAC) inhibitors, including Trichostatin A, were scored as top hits (Fig. 5c and Supplementary Data 4), whose gene expression responses in CMap database were negatively enriched for the let-7i-5p signature, leading to the hypothesis that inhibiting HDAC could decrease let-7i-5p expression. This was again experimentally verified by treating K562 cells with Trichostatin A, which led to a significant downregulation of let-7i, with no effect on miR-146b-5p (Fig. 5e, g). MK-2206 and cyclopamine that target other pathways had no effect on the expression of let-7i-5p (Fig. 5e, h). Trichostatin A did not alter the level of primary let-7i transcript, suggesting post-transcriptional control on let-7i-5p expression (Fig. 5f). Taken together, the above results support that informative regulatory relationships for miRNA expression could be revealed through single-cell miRNA and mRNA co-profiling.

## Discussion

In recent years, substantial progress has been made in single-cell genomics and previous studies reported on the separation of DNA and RNA for co-measurement of pre-transcriptional alterations (e.g., DNA methylome) and mRNA gene expression.

Here, we achieved co-analysis of post-transcriptional regulators (miRNAs) and the mRNA transcriptome in single cells. This was realized with a half-cell genomics approach, which overcomes the challenge to separate and capture small RNAs from mRNA without introducing material loss and technical variation[35]. We evaluated the validity of sequencing half-cell RNA materials to measure single-cell miRNA or mRNA expression profiles, and further combined half-cell miRNA and mRNA profiles from the same single cells to perform multi-omic analysis. Having the genome-wide miRNA and mRNA profiles from the same single cells permits direct assessment of whether intercellular mRNA transcriptomic heterogeneity is in part shaped by the intercellular variability of the corresponding miRNA regulators. We observed that several abundantly expressed miRNAs, including miR-92a-3p, miR-125a-5p and miR-26a-5p, were more inversely correlated with its predicted targets than non-targets, supporting the above notion. Predicted targets with abundant expression levels were more strongly anti-correlated with the miRNA than predicted targets with lower expression. Why was a stronger anti-correlation observed only for predicted targets of high abundance? One possibility is that the higher technical noise (on log2-transformed data) for genes with low expression levels resulted in the lower degree of statistical significance even if there were correlation between miRNA and targets. Another possibility is that protein-coding genes with low expression levels are more prone to regulation by miRNAs of both high and low abundance, diluting the correlation with any given single miRNA. In contrast, protein-coding genes with high expression levels are susceptible to regulation only by abundantly expressed miRNAs, leading to the preservation of correlation with the corresponding miRNA. This second possibility is consistent with recent findings that support the efficacy of miRNA-mediated regulation being a function of the ratio between miRNAs and targets[49–52]. Overall, our experiments opened up the opportunity to interrogate single cells for post-transcriptional regulation of protein-coding genes to modulate mRNA cell-to-cell variability.

In addition to assessing miRNA-based regulation of mRNA heterogeneity, the availability of paired half-cell mRNA and miRNA profiles allowed us to derive informative and verifiable hypotheses on regulatory relationships between cellular pathways and miRNA expression that may contribute to intercellular heterogeneity of miRNA expression. Notably, the expression of miR-146b-5p could be regulated by an AKT inhibitor, which is well recognized to control protein translation. This regulation seems to occur at the level of miRNA maturation, because the primary transcript was reduced yet the mature miRNA was increased. This effect cannot be explained by previous findings of AKT controlling AGO2[53], which would predict similar responses to

AKT inhibition by all miRNAs. The lack of effect of the AKT inhibitor on other miRNAs (miR-92a-3p and let-7i-5p) and the decrease of primary miR-146b transcript by AKT inhibition indicate that this modulation occurs at a separate step from AGO2 regulation. The exact mechanism requires further studies to elucidate. Similarly, we demonstrated that the let-7 family member let-7i-5p could be downregulated by HDAC inhibition. HDAC inhibition is often associated with activation of gene transcription, which did not seem to occur with let-7i, as the primary transcript of let-7i was not altered, again supporting the existence of post-transcriptional regulation of let-7i-5p by HDAC. It is thus interesting that post-transcriptional regulation of both mRNA and miRNA can be revealed on the single-cell level through co-analysis of mRNA and miRNA transcriptomes.

While our study demonstrated the feasibility and utility of obtaining paired half-cell mRNA and miRNA profiles, our current approach has limited throughput in terms of the number of cells per run. Integrating the protocols demonstrated in this work in microfluidic systems could potentially improve on the throughput. Additionally, while this study focused on miRNAs, the half-cell small RNA sequencing data also contain information of other small RNA species, such as tRNA fragments that have been shown to be functionally important in human cancers and other diseases[54–56]. In addition, small RNAs mappable to other genomic regions could be readily detected in our half-cell profiles (Supplementary Data 5). We envision that further development and application of half-cell co-profiling of small RNAs and mRNAs could lead to further mechanistic insights into small RNA biology.

## Methods

**Cell culture.** K562 and MCF-7 cell lines were obtained from American Type Culture Collection (ATCC). K562 cells were maintained in RPMI 1640 media supplemented with 10% heat-inactivated fetal bovine serum (FBS, Life Technologies # 16000044) at a cell density between $2 \times 10^5$ and $2 \times 10^6$ ml$^{-1}$ at 37 °C with 5% $CO_2$. The murine BaF3 cells were from our laboratory stock and cultured in RPMI 1640 medium containing 10% FBS, 1% of 100× Penicillin–Streptomycin–Glutamine (Life Technologies # 10378016) and 3 ng ml$^{-1}$ of recombinant murine IL-3 (Peprotech #213-13). MCF-7 cells were cultured in DMEM media (Life Technologies, #11995065) with 10% FBS and 1% of Penicillin–Streptomycin–Glutamine. K562 cells were tested and showed sensitivity to Imatinib. BaF3 cells were tested and showed sensitivity toward IL-3 withdrawal. MCF-7 cells were tested and showed sensitivity toward estrogen receptor inhibition. Cells were acquired mycoplasma-free, but have not been specifically tested for mycoplasma in this study.

For deriving K562 single-cell clones, single K562 cells were hand-picked under microscope (see Cell Lysis for details) and maintained in single wells of a 96-well plate. Visual inspection was performed to ensure that only a single cell was present per well. Expanded single-cell clones were harvested after ~20 days from the initial single-cell plating and further expanded for total RNA preparation.

For chemical treatment experiments, K562 cells were plated at 50,000 cells per well in a 96-well plate. Cells were treated with 10 μM of cyclopamine, 10 μM MK-2206, 1 μM Trichostatin A, 5 μM Afuresertib (AFU) or 10 μM Ipatasertib (IPA). All chemicals were from Sigma Aldrich, except for AFU and IPA which were from Selleckchem (GSK2110183 and GDC-0068). Chemicals were dissolved in DMSO, and DMSO was added to the control group. Cells were harvested after 24 h for total RNA extraction.

**Oligonucleotides and primers.** Sequences and modifications are available in Supplementary Data 6.

**Cell lysis and pre-treatment.** Cells were washed once with cold PBS (Corning, #21-031-CV) and diluted to< 1000 cells ml$^{-1}$. Prior to picking single cells, 10 μl of lysis buffer (0.25% Triton X-100, Sigma #T8787, 4 units (U) recombinant RNase Inhibitor, Takara #2313 A) was deposited in each PCR tube. All single-cell experiments were performed using 8-strip PCR tubes (USA Scientific #1402–4700) and 10 μl filter tips (Denville #P1096-FR). To ensure single-cell resolution, we performed most experiments by hand-picking of single cells under microscope. The diluted cell mixture was placed on the cover of a petridish, and single cells without close neighbors were manually picked using a 10 μl pipette, with each cell contained within less than 1 μl of volume. Single cells were then added into the lysis buffer. The single-cell lysates were stored at −80 °C. After thawing, the lysates were incubated at 72 °C for 20 min or 75 °C for 5 min to release small RNAs for reaction.

The lysate was then split into two ~5 μl halves, with each half transferred into an empty PCR tube.

**Small RNA library preparation and sequencing.** Preparation of the 3′ adaptor: 5′-phosphorylated 3′ adaptor oligonucleotides were ordered from IDT. Pre-adenylation of 5′-phosphorylated 3′ adaptor was performed by using the 5′ DNA adenylation kit (NEB, #E2610S) and purified using the Nucleotide Removal kit (Qiagen, #28304), both following the manufacturer's protocol. The concentration of the pre-adenylated adaptor was estimated by analysis on an 15% denaturing polyacrylamide gel (made with American Bioanalytical Sequel NE, #AB13021, AB13022) and comparison of band intensity to known quantities of synthesized oligonucletides. The pre-adenylated 3′-adaptor (referred to below as 3′ adaptor) was stored at −80 °C in aliquots. 5′ Adaptor was ordered from Dharmacon. All other oligonucleotides were from IDT. Purification of oligonucleotides was performed by the synthesis companies.

Small RNA library preparation: Single-cell or half-cell lysate (5 μl) was subjected to 3′ adaptor ligation by adding 5.24 μl of 3′ adapter ligation reaction mixture (1 pmol 3′ adapter, 1.66 μl 50% PEG 8000, NEB #M0373S, 250 U T4 RNA Ligase 2 truncated KQ, NEB #M0373S, 0.83 μl 10 × T4 RNA ligase buffer, NEB #M0373S, 20 U recombinant RNase Inhibitor, Takara, #2313 A) and the reaction was incubated at 30 °C for 6 h followed by 4 °C for 10 h. Next, 5 μl adaptor-digestion mixture was added (10 pmol RT primer, 5 U Lambda exonuclease, NEB #0262 S, 5 U 5′ deadenylase, NEB #M0331S) and the reaction was incubated at 30 °C for 15 min followed by 37 °C for 15 min. Next, 5 μl 5′ adapter ligation reaction mixture was added (10 pmol 5′ adapter oligo, 10 U T4 RNA ligase 1, Thermo Fisher #EL0021, 1 μl T4 RNA ligase buffer, Thermo Fisher #EL0021, 2 μl 50% PEG 8000, NEB #M0373S) and incubated at 37 °C for 1 h. Reverse transcription reaction was performed in three steps using M-MLV reverse transcriptase (RT) (Invitrogen #28025013). First, 5 μl of RT reaction mix was added (10 pmol RT primer) and the reaction was incubated at 65 °C for 5 min. Second, the mix of 2.1 μl 5× First-Strand buffer (Invitrogen, #28025013), 0.3 μl H₂O and 0.8 μl of 0.1 M DTT (Invitrogen #28025013) was added and incubated at 42 °C for 30 min. Third, 3.3 μl of RT reaction was added (0.6ul M-MLV reverse transcriptase, 1 μl 5× First-Strand buffer, Invitrogen #28025013, 0.3 μl H₂O, 0.8 μl of 0.1 M DTT, Invitrogen #28025013 and 0.6 μl 10 mM dNTP mix, Invitrogen #18427088) and the reaction was incubated at 42 °C for 2 h. The first PCR amplification was carried out by adding 35 μl of the reagents (3.5 μl 10×ThermoPol Reaction Buffer, NEB #M0267S, 0.7 μl 10 mM dNTP mix Invitrogen #18427088, 35 pmol RT primer oligo, 35 pmol PCR primer oligo, 0.5 μl Taq DNA Polymerase NEB #M0267S, 23.3 μl H₂O) and incubating at 98 °C for 30 s followed by 13 cycles of 98 °C for 10 s, 60 °C for 30 s and 72 °C for 30 s and a final incubation at 72 °C for 5 min. Next, 1 μl of the amplified product was transferred to a fresh tube and to 25 μl of second PCR reaction (consisting of 10 μM indexed primer, 10 μM 5′ Illumina PCR primer, 2.5 μl 10× ThermoPol Reaction buffer, 0.5 μl Taq DNA Polymerase NEB #M0267S, 0.5 μl 10 mM dNTP mix, Invitrogen #18427088), followed by a 30 s incubation at 98 °C, 13 cycles of 98 °C for 10 s, 67 °C for 30 s, and 72 °C for 30 s and a final incubation at 72 °C for 5 min. PCR products of ~140 bp were gel-purified after the second PCR on an 8% non-denaturing acrylamide gel. Gels were prepared by using Protein Gel Mix (American Bioanalytical #AB00283). Samples were prepared by adding 1/5 volume of 6X Gel loading dye purple (NEB #B7024S). Electrophoresis was performed at 140 V for 80 min. After electrophoresis, the gel was stained in 15 ml water containing 1.5 μl GelStar Nucleic Acid Gel Stain (Lonza #50535) for 15 min. Bands were cut, eluded in 0.3 M sodium chloride overnight and precipitated.

All barcoded small RNA libraries were sequenced on an Illumina HiSeq 2000 instrument. Alternatively, for Luminex-based detection, the second PCR was performed with a 5′ biotinylated primer, and detected on the Luminex platform as previously published[57].

**SMART-Seq mRNA profiling library preparation and sequencing.** We followed the manufacturer's protocol of SMART-Seq v4 Ultra Low Input RNA kit for sequencing (Clontech #634889). Then, Nextera DNA Library Preparation Kit (Illumina #FC-131-1096) was used to fragment and prepare the sequencing library. The libraries were sequenced on an Illumina HiSeq 2000 instruments with paired end 100 base reads.

**Massively parallel single-cell 3′-enriched RNAseq.** Massively parallel single-cell 3′-end mRNA sequencing was performed by adopting the DropSeq chemistry procedure[37] to microwell arrays, modified from those demonstrated before[38,39]. Microwell arrays were fabricated in polydimethylsiloxane (PDMS) using standard soft lithography. Each array consists of 15,000–50,000 microwells, and microwells are sized 55 micron in diameter and 45 μm in depth to ensure capture of only one barcoded bead with poly-T-containing probe in each well by size exclusion. Cells were loaded into the device such that only 5–10% of the wells were loaded with single cells to prevent cell doublets and verified by microscopy before proceeding further. Once cell loading was completed, beads were loaded into the array at high density to ensure that each well receives a single bead (>95% bead occupancy). Following bead loading, lysis buffer was introduced into the array, and the array was then immediately sealed by introducing fluorinated oil. The cell lysates and beads were then incubated for an hour to capture the released mRNA

molecules onto barcoded beads. Beads were then removed from the devices by centrifuging the devices upside down to push the beads out of microwells and flushing them out into an Eppendorf tube. The beads are then processed with subsequent reverse transcription, exonuclease treatment, amplification, and library preparation as in the Dropseq protocol, and sequenced using paired end sequencing. After sequencing, read alignment and generation of gene expression matrices were performed as described in the DropSeq protocols[42].

**Small RNA sequence analysis**. Small RNA reads were analyzed by a custom perl pipeline that has been described[58]. This pipeline was based on the miRDeep2 package[56]. Briefly, after removing adaptor sequences, collapsing reads with the same sequence, and removal of reads with less than 15 bases in length, small RNA reads were mapped to miRNA precursors from miRBASE release 21 using miRDeep2, allowing zero mismatches. Mapped reads were quantified by miRDeep2. In addition, reads were also mapped to several categories of known RNA species sequentially, using bowtie2. Reads were first mapped to human miRNA precursors, with unmapped reads used in the next step to map to snoRNAs. Similar sequential mapping was performed for snRNA, scRNA, tRNA, rRNA, and piRNAs. Unmapped reads left from the above mapping were also mapped to the genome. However, we found that there are often sequence variants of the ligation adaptors with perfect or near perfect mapping to the genome, thus creating artefacts. We thus did not utilize the mapping to the genome to quantify the mapping results. Overall, among the 19 half cells, total reads that map to miRNA account for 30.18 $\pm$ 7.62% (mean $\pm$ standard deviation, $N = 19$ cells; same for numbers below) of all reads mapping to the above mentioned RNA species. Reads that map to snoRNA, snRNA, scRNA, tRNA, rRNA, and piRNA represent 8.23 $\pm$ 2.41%, 1.02 $\pm$ 0.33%, 2.17 $\pm$ 1.07%, 6.11 $\pm$ 0.88%, 52.28 $\pm$ 9.27%, and ~0.01% respectively.

Normalization of miRNA reads were performed by dividing the reads quantified by miRDeep2 with total miRNA reads quantified by bowtie2. This is because some reads were counted multiple times when miRDeep2 assigned them to miRNA precursors, especially for miRNAs that map to multiple genomic loci. The resultant data reflect the fraction of a given miRNA within all miRNAs, which we refer to as miRNA expression or miRNA levels in figures. The validity of using this fraction-based data was supported by single-cell qRT-PCR validation. Normalized data were further applied with a minimal expression level of 10e−4 (i.e., all values lower than this threshold were set to this level), and log2-transformed for further analyses.

**RNAseq data analysis**. RNAseq data from the half-cell profiles were processed using GenePattern (v3) software. Specifically, K562 RNAseq data were aligned using Tophat to hg19 genome assembly, using gtf files based on UCSC gene definition downloaded from the GenePattern server. Aligned reads were quantified using the CuffDiff program, by using UCSC gtf definition for hg19, to obtain RPKM values. Among the 19 half K562 cell profiles, the number of genes detected is 11,209 $\pm$ 592 per half-cell ($N = 19$ cells). Data were applied with a minimal expression level of 10e−3, and log2-transformed for further analyses.

To identify similar groups of cells, consensus clustering was performed using matlab codes. Specifically, for each iteration, 80% of samples were randomly sampled, and subjected to hierarchical clustering. Hierarchical clustering was performed with average linkage and Pearson correlation metrics after row-based centering and normalization (subtracting mean of each row and dividing by standard deviation of each row). A total of 50 iterations was performed, and consensus clusters of 2 to 5 were manually examined. Three clusters were selected based on the clustering results for further analyses.

To identify differential gene expression between clusters, permutation was performed to shuffle sample labels, with a total of 5000 iterations, and using t-test score as a metric. Nominal P- values were obtained by quantifying the fraction of permutations that reached a more extreme t-test score. False discovery rates were then computed based on the nominal P-values using the Benjamini–Hochberg method. For 19 K562 half-cell data, the differential gene expression was defined with FDR < 0.1. For ~400 K562 single cells, the differential gene expression was defined with FDR < 0.05. Enrichment of pathways was performed by DAVID (v6.8) analysis with the corresponding groups of differentially expressed genes.

**Co-analysis of miRNA and mRNA**. To analyze the relationship between miRNA and miRNA targets, mRNA genes were ranked by the Pearson correlation to miRNA expression among the 19 K562 half cells. Initial analyses were performed using gene set enrichment analysis (GSEA v2.0), using miRNA-target gene set defined in MSigDB, with 1000 or more permutations on gene labels, and with Pearson correlation as weights during the enrichment analyses. In addition, miRNA-target genes defined in TargetScan release 7.1 were also examined, using the same enrichment algorithm as in GSEA.

For examining miRNA-target relationships using cumulative distribution functions, the predicted miRNA targets were obtained from TargetScan release 7.1, and filtered to eliminate predicted targets with poor total context scores. Specifically, we only retained predicted targets that had total context scores < −0.1. Targets were further categorized according to mean log2 expression (measured in RPKM). P-values were calculated based on the Kolmogorov–Smirnov test, using matlab's kstest2 function.

For any given miRNA, the rank list of mRNA according to the correlation was also used to query MSigDB for potential pathways or gene sets. To query the Connectivity map (cMap, build 2), the ranked mRNAs were separated into positively correlated genes and negatively correlated genes, with a cutoff of 0.45 or −0.45 for the correlation score. These two lists of gene symbols were then mapped to the probe names of U133A chip to query CMap.

**Single-cell quantitative real time RT-PCR**. Single-cell qRT-PCR was performed as detailed below following a published study[19] with modifications. For preparation of cell lysate for qRT-PCR, prior to picking single cells, 4 µl of lysis buffer (1% Triton X-100, 4 U recombinant RNase Inhibitor, Takara #2313 A) and (2e−19 mole of synthetic hsa-miR-371-5p, Ambion) was deposited in each PCR tube (TempAssure PCR 8-Stripes, USA Scientific #1402–4700). The spiked-in hsa-miR-371-5p permitted final data normalization with a miRNA not expressed endogenously in K562 cells. K562 cells were first washed once with cold PBS. K562 single cells were manually picked by using a 10 µl pipette in 1 µl volume and added to lysis buffer. Cell lysate was stored in −80 °C, thawed and incubated at 75 °C for 5 min and subjected to miScript RT reactions (Qiagen, by adding 2 µl 5× miSript HiSpec buffer, 1 µl 10× nucleotides, 1 µl miScript RT enzyme mix). RT product was diluted 20 times (v/v) with water, and 1 µl diluted RT product was used for Pre-PCR reaction in 10 µl total volume with Power SYBR-green PCR master mix (Applied Biosystems) for miRNAs. For a given experiment, all relevant miRNA-specific primers (2 pmol each), as well as that of the spike-in control were added. See Supplementary Data 6 for specific primers used. qPCR reaction was then performed after pre-PCR and the dilution of pre-PCR product 20 times with water (v/v). The total volume for the reaction was 10 µl (5 µl Power SYBR-green PCR master mix, 2 pmol miScript universal primer, 2 pmol miRNA-specific primer, 2 µl diluted pre-PCR product). The expression of each miRNA was quantified by comparative Ct method ($2^{-\Delta Ct}$), using the detected levels of hsa-miR-371-5p as a control.

**RNA extraction and bulk population qRT-PCR**. Total RNA was extracted using the TRIzol Reagent (Life Technologies #15596018) following the manufacture's protocol. For qRT-PCR of bulk population samples, 10 ng of total RNA was used for each biological sample. U6 small RNA was used as a control for qRT-PCR of miRNAs. All RT, qPCR and data analysis details were following the procedure in above sections.

**Statistical analysis**. Student's t-test (unequal variance, two-tailed) was used in analysis of chemical treatment experiments. Number of replicates were determined after an initial experiment with three biological replicates to gauge the data variation. For analysis of correlation between miRNAs and its predicated targets, Kolmogorov–Smirnov test was used (two-sided) through Matlab kstest2 function. Statistics of gene set enrichment analysis was obtained from the GSEA program output, using the familywise-error rate. For evaluating the variation of miR-146 with AKT inhibitor treatment, P-value was calculated using custom matlab codes by performing random permutations. Specifically, for both control and AKT inhibitor groups, 15 samples per group were randomly drawn to calculate standard deviations. A total of 1000 permutations were performed. P-value was calculated as the probability of rejecting the hypothesis that the AKT inhibitor group has lower variation than the control group, by dividing the number of random instances, in which the standard deviation of the AKT inhibitor group is larger or equal to that of the control group, by the total number of random instances. P-value of 0.006 was observed. Similar significant results were obtained with random sample size from 13 to 18.

**Reporting summary**. Further information on experimental design is available in the Nature Research Reporting Summary linked to this article.

**Code availability**. The custom code used for data analysis is available as Supplementary Software 1.

## Data availability

Next-generation sequencing data have been deposited to GEO under series GSE114071. A reporting summary for this Article is available as a Supplementary Information file. All other data supporting the findings of this study are available from the corresponding authors on reasonable request.

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

## Acknowledgements

This work was supported in part by NIH grants R01CA149109 (to J.L.), R01GM116855 (to Y.D. and J.L.), R21CA177393 (to R.F.), U54CA193461 (to R.F.), U54CA209992 (Sub-Project ID: 7297 to R.F.), Yale Cancer Center Pilot Grant (to R.F.), Sackler Institute Seed Grant (to R.F.), Connecticut RMRF grant 15-RMB-YALE-06 (to J.L.), and National Science Foundation CAREER Award CBET-1351443 (to R.F.). Services provided by the NIDDK-supported Yale Cooperative Center of Excellence in Hematology (U54DK106857) assisted this study.

## Author contributions

R.F. and J.L. conceived the study and supervised the work. N.W. established the single and half-cell small RNA profiling method. N.W. and Z.C. generated half-cell mRNA profiles. N.W., J.Z and YLiu generated single/half-cell miRNA profiles, performed single/half-cell qRT-PCR, and performed experiments on drug treatment. B.D., M.K., and Z.C. established, obtained and analyzed data for high-throughput single-cell RNAseq. J.X.-F., Y.-C.L. and D.K. contributed to the drug treatment experiments. M.Z., C.R., and J.C. contributed to cell acquisition and subsequent experiments. J.L. and Y.D. analyzed data from single-cell profiles. N.W., R.F., and J.L. wrote the manuscript.

## Additional information

**Competing interests:** Rong Fan is on the Scientific Advisory Boards of IsoPlexis, Bio-Techne, and Singleron Biotechnologies. The remaining authors declare no competing interests.

