## [Peer Review File · Nature Communications]

Reviewers' comments:

Reviewer #1 (Remarks to the Author):

Wang et al. report the first co-measurement of miRNA and mRNA levels from the same single cells using a half-cell approach. They apply their method to only 19 single cells of the K562 cancer cell line. The generated data is used to explore relationships between miRNA and target mRNA expression levels. Furthermore, the authors use their data to suggest regulatory relationships controlling miRNA expression, which are validated using complementary bulk methods. However, I am not entirely sure that this work would significantly increase our understanding of the consequences of miRNA variability and I do not believe that the study meets the high-level standard required for publication in Nature communications. Though, regarding acceptance or rejection I also leave the decision to the editor as well as the reports and opinions from the other referees.

Major issues:

- 1) In the introduction, the authors neglect to cover previous work examining co-profiling of miRNA and mRNA levels in bulk cell populations^{1–4}. This kind of work is relevant to this study as it uses a different source of miRNA and mRNA variability which could be used for some of the analysis done by the authors on their single cell data.
- 2) The number of single cells used for the various experiments is very low when taking into account the throughput of modern single cells analysis tools. Specifically, I would expect an order of magnitude more single cells used for the majority of the co-profiling analysis while the authors use only 19 cells. Furthermore, when assessing the reproducibility of their methods, the authors measure the correlation of miRNA and mRNA expression profiles from only a single cell. It would be more convincing to present such analysis performed on multiple single cells.
- 3) Given the previously established methods for mRNA and miRNA sequencing used by the authors, they do not present any major technological advancement beyond the optimization of the usage of half-cells.
- 4) The authors should not conclude that the half cells from the same cell are more similar than the half cells from other cells when they have only one cell used for the half cells from the same cell. Multiple pairs and a proper statistical analysis should be applied. Furthermore, in order to make the claim of half-cells from the same cell being more similar than half cells from different cells, as shown in Fig. S4, the authors need multiple pairs of half-cells from the same single cells.
- 5) I would expect the authors to compare their single cell measurements for miRNA and mRNA levels to bulk measurements performed on the same cell line in the same conditions.
- 6) The authors mention the dataset from Vaz et al. 2010⁵ when comparing the expression levels of miR-92a. It would be appropriate to show how their mean miRNA expression levels, computed from the single cell measurements, compare to the complete dataset. A scatter plot and a correlation analysis should be presented. This should reveal how well the author's miRNA measurements agree with external data.
- 7) When validating the variability in miRNA expression using qRT-PCR (Fig. 2c), a direct comparison between the qRT-PCR and the half cell miRNA sequencing results is lacking. Furthermore, I would expect the qRT-PCR to be performed on half cells as well, for a valid comparison. Moreover, a comparison of the expression means should also be performed.
- 8) It is not clear to me how the analysis performed in Fig. 2d shows that the variations in miRNA expression could not be fully attributed to genetic differences but instead associated with non-genetic cellular heterogeneity. The interpretation of the results should be clarified in the text.
- 9) The quality check for the RNA-seq data from 400 cells performed by the authors should also include a direct comparison of gene expression levels and variability with the data generated by InDrop.
- 10) It is not clear to me how the analysis in Fig. S10 was performed and how it shows a correlation between transcriptional clusters and top ranked miRNAs. On what values was the clustering performed? Why were specific clusters attributed to the annotated miRNAs? How does the heatmap convey the desired message?

11) "indicating the possibility for the top-ranked miRNAs to not only regulate specific targets but also influence global transcriptional phenotype in single cells." . The authors imply causation where only a correlation is observed.

12) "For let-7a-5p and let-7i-5p, BCL2L1 decreased as let-7 expression increased at the high end of miRNA expression, suggesting a possible role for let- 7a/let-7i to alter the level of its target mRNA BCL2L1. Such correlation was not seen at the low end of let-7a/let-7i expression levels, as expected." For let-7a-5p one can also observe that BCL2L1 increased as let-7 expression increased at the low end of miRNA expression. The authors seem to report their interpretation of the data in a biased manner.

13) The author's results support the notion that a negative relationship between miRNAs and their targets can be observed in their half cell data. However, in order for them to claim a functional relationship, as they do, it is not sufficient to associate these relationships with clustered transcriptional programs, in my opinion. A functional relationship could be shown by experimental association of observed relationships with some measurable phenotype such as apoptosis markers, cell cycle phase or differentiation markers.

14) The authors use their data to explore regulatory relationships controlling miRNA expression. It is true that this exploratory analysis triggered the identification of a regulatory relationship for miR-146-5p, followed by verification on bulk cultures as opposed to single cells. However, comparative analysis of bulk miRNA and mRNA expression data across cell lines or a single cell line in different conditions could be used as an alternative starting point. The use of cell to cell variability in miRNA expression levels could, in my opinion, be replaced by any other source of miRNA expression variability. Therefore, I fail to note a significant advantage of the author's methods in uncovering such regulatory relationships.

15) "This was realized with a halfcell genomics approach, which overcomes the challenge to separate and capture small RNAs from mRNA without introducing material loss and technical variation." A reference or supplementary data for these challenges should be included since this claim poses the main motivation for using the author's methods as opposed to potential alternatives.

Minor issues:

- 1) In Fig. S1 x and y axis labels are inconsistent.
- 2) Given the optimization of lysis conditions presented in Fig S1, it is not clear which method was used for the generation of Fig 2a.
- 3) "We further examined the validity of this half-cell approach using different cell types and a different measurement techniques.". However, the authors only show one different cell type with one different measurement technique.
- 4) In Fig. S5 and S6 the usage of a divergent colormap is not justified. A sequential colormap should be used. Moreover, the usage of a pure blue color for self-correlations while the lower correlations use blue hues is a very poor choice.
- 5) In Fig. S5 there is one cell that seems to have higher correlations with all other cells, how can this be explained?
- 6) "Comparing intercellular variability across miRNAs, both miR-146b-5p and let-7i-5p showed higher variability than miRNAs with similar expression levels.". The authors should accompany these claims with statistical analysis.
- 7) MiR-let-7a is missing from Fig. 2d.
- 8) In Fig. 4f the x-axis should also be labeled and not just explained in the legend.
- 9) In the context of Fig. 4f, the correlation coefficient and the p-value for the correlation of miR-125a-5p with BCL2L1 should be reported.

References:

1. Gennarino, V. A. et al. MicroRNA target prediction by expression analysis of host genes. *Genome Res.* 19, 481–490 (2009).
2. Naifang, S., Minping, Q. & Minghua, D. Integrative Approaches for microRNA Target Prediction: Combining Sequence Information and the Paired mRNA and miRNA Expression Profiles. *Curr.*

Bioinforma. 8, 37–45 (2013).

3. Cho, S. et al. miRGator v3.0: a microRNA portal for deep sequencing, expression profiling and mRNA targeting. *Nucleic Acids Res.* 41, D252–D257 (2013).

4. Giles, C. B., Girija-Devi, R., Dozmorov, M. G. & Wren, J. D. mirCoX: a database of miRNA-mRNA expression correlations derived from RNA-seq meta-analysis. *BMC Bioinformatics* 14, S17 (2013).

5. Vaz, C. et al. Analysis of microRNA transcriptome by deep sequencing of small RNA libraries of peripheral blood. *BMC Genomics* 11, 288 (2010).

Reviewer #2 (Remarks to the Author):

Wang et al. *Nat. Comm.* 2017

The manuscript is a follow-up on the authors' previous paper from 2017 (ref. 2 of this manuscript), in which the authors developed a small RNA sequencing method for single cells. Unlike established methods, where genomic DNA and mRNA are co-analyzed, they present an innovative technology for parallel sequencing of both mRNA and small RNA of the same single cell, in order to gain new insights into post-transcriptional modulation of non-genetic intercellular heterogeneity by miRNAs and how miRNA variations might be affected by protein-coding genes. To this end, the authors split cell lysates into halves and perform the two incompatible sequencing procedures (their own small RNA sequencing technology and the commercial SMART-Seq protocol for mRNAs) separately on these halves. They first validate this approach and then apply it to identify a yet unknown regulation of miR-146b-5p by the AKT pathway.

Major criticisms and concerns

1. Novelty, methods and foundation of claims.

Parallel analysis of miRNA and mRNA has not been done in the same single cells has not been published so far. In this regard the approach presented in the paper is novel. However, the method is entirely based on the half-cell approach and I am missing data on the reproducibility of the splitting method, as the whole technology rests on the assumption that miRNAs and mRNAs are homogeneously distributed between both halves of a single cell.

a. The authors are showing the correlation between two halves of the same cell only for three single cells in the paper (in Figure 2a and Supplementary Figures 1 and 2). I am certain they tested this on more than three cells. I would like to know how many times these experiments were replicated and to see the average correlation of halves from the same cell as proof that the approach is valid and that the high correlation they observed was not random.

b. Heterogeneity may have several sources. For a general reproduction of the data by the readership, important information on the methods are missing. For example, what kind of PCR tubes and pipette tips were the authors using for their experiments? This is not mentioned. I think this is relevant here, since nucleic acids tend to stick to normal, uncoated plastic. When working with bulk RNA from a large number of cells this does not matter, of course, but it is a major issue when working with single cell material, because it may cause significant loss of said material.

c. Another way to address the origin of heterogeneity (true cell-to-cell heterogeneity or technical noise) is the use of spike-in experiments. This could be done for instance by conducting a series of spike-in experiments with artificial miRNA molecules introduced at single-cell levels in defined amounts. Provided that half-cell sampling approach is unbiased and miRNA amplification efficient across the physiological range of miRNA expression the expected outcome would be: (i) equal amount of spike-ins detected in both halves of the single-cell lysate (applies to spike-in representing all – high, medium and low – miRNA expression levels) and (ii) stable and reproducible quantities or reads corresponding to each spiked miRNA, irrespectively of the initial amount of the molecules introduced in the lysate.

2. Methodological and functional validation

In general, I found the functional validation experiments elegant and interesting. However, the paper would definitely benefit from further validation, i.e. by testing the detected relationships in independent model system (e.g. multiple cells lines) or primary cells. Basically, the authors have used only one cell line (K562) and validation of the general approach with at least additional cell lines (if not clinical samples for the general aspects, such as feasibility with "real life" miRNA and mRNA levels) would be important.

3. Graphical representation of data.

It is repeatedly not clear what type of data was presented in the figures. For clear and complete understanding of the paper comprehensive explanation of the data presented in the figures is essential. For instance, a term "normalized and log₂-transformed miRNA expression levels" does not explain precisely what kind of data was depicted in the plot.

4. Sequencing performance of the new workflow.

The manuscript does not state how the success of single-cell miRNA sequencing was defined. No quality metrics were provided addressing the performance of either single-cell miRNA or mRNA sequencing – e.g. the amount/proportion of reads mapping to other portions of the genome (different to expected miRNA and mRNA sequences), amount/proportion of non-mappable reads, etc.

Minor criticism:

There are numerous syntax errors or missing words throughout the manuscript. Many of these errors reduce legibility and should be corrected.

1. The y-axes of both plots of Supplementary Figure 1 have typing errors in their labels: "loh₂" instead of "log₂".
2. There is a typing error in the x-axis label of Supplementary Figure 2.
3. Some graphs are not precisely labeled, e.g. "Log₂ (miRNA Level)". What is the miRNA level exactly? In some graphs the labels actually say "FPKM", but often the units are missing.
4. Supplementary Figure 7: I do not consider library size and number of expressed genes suitable measures of sequencing quality. Please, add average PHRED score distribution across reads.
5. Supplementary Figure 8 needs to be improved for better legibility. The genes need to be sorted in the same way and need to have the same colors in both panels. Right now, it takes too much time to find the same gene in both panels, in order to compare its expression.
6. Figure 4f should have a label for the x-axes. The explanation in the legends is not enough.

Reviewer #3 (Remarks to the Author):

The manuscript by N. Wang and colleagues refers to co-sequencing of both mRNAs and microRNAs from the same single cell by implementing a half-cell approach. The authors developed a strategy to co-profile miRNAs and mRNAs from a single-cell lysate split into two half-cell fractions. The lysates then underwent either miRNA or mRNA transcriptome sequencing.

The manuscript is overall well written. As far as I know, this is the first attempt to combine genome-wide profiling of both mRNA and post-transcriptional regulators from the same single cell and I think the study will be of great interest to the community.

My major claims, however, involve the comparison between the two half-cells, both sequenced for miRNAs or mRNAs, and the whole miRNA or mRNA single-cell sequencing.

The entire work is based on the assumption that the half-cell sequencing procedure is working

properly based on only one single cell experiment (figure 2a and 3a). Since sampling molecules from a fixed volume and sampling from half of this volume is in principle very different, I think that in order to show that one half-cell is really representative of the other half, the authors should show results for more than one single cell with both halves sequenced for miRNAs or mRNAs. Said differently, they should show that the half-cell sequencing results presented in Figures 2a and 3a are reproducible in terms of correlation (and PCA analysis for miRNAs (suppl. Fig.4)) between the two halves for more than one cell (that is, for a number of cells comparable to the 19 presented) for both mRNA and miRNAs.

Minor comments:

With respect to Figure 2a-b, why is the number of annotated miRNAs different in the two panels? If so, the authors should show panel (a) with the same annotated miRNAs as panel (b) (or at least justify the difference between the two).

Few typos:

In Figures 2b "Standard deviation" is written SD while in figure 3b is "Std";

In the text the authors always refer to Pearson coefficient r while in the figures they show R^2

Summary:

We thank the reviewers for valuable comments that helped to improve this manuscript. We have included multiple new experiments and data to address reviewers' concerns, including those on the reproducibility of lysate splitting, and to examine miRNA gene regulation in other cells.

We have also made changes to the manuscript to comply with journal format requirements, including but not limited to the shortening of subsection titles.

We have highlighted added new text in red in the main manuscript file.

Point-by-point response to reviewers:

Reviewer #1

Overall Comment: Wang et al. report the first co-measurement of miRNA and mRNA levels from the same single cells using a half-cell approach. They apply their method to only 19 single cells of the K562 cancer cell line. The generated data is used to explore relationships between miRNA and target mRNA expression levels. Furthermore, the authors use their data to suggest regulatory relationships controlling miRNA expression, which are validated using complementary bulk methods.

However, I am not entirely sure that this work would significantly increase our understanding of the consequences of miRNA variability and I do not believe that the study meets the high-level standard required for publication in Nature communications. Though, regarding acceptance or rejection I also leave the decision to the editor as well as the reports and opinions from the other referees.

Response: We thank the reviewer for recognizing that this study is the first co-measurement of miRNA and mRNA levels from the same single cells and the data enabled us to reveal the fundamental relationship between miRNA and mRNA expression levels, which seem to us innovative and important. Therefore, we respectfully disagree with the reviewer's other comment regarding the significance and novelty. In addition, our work is not only about the understanding of the consequences of miRNA variability. Instead it devised a strategy and demonstrated the technical feasibility of profiling miRNAs and mRNAs from the same single cells; our work demonstrates the correlative relationship between single-cell miRNA variability and miRNA targets on the genomic scale, which has never been possible to examine before; our work reveals new relationships that control miRNA expression. In addition, our new data (see **Supplementary Figure 11b** and response to comments 14 below) further confirms that AKT activity controls the variability of miRNA expression in single cells (for miR-146b), which has never been demonstrated or explored before.

Comment 1: In the introduction, the authors neglect to cover previous work examining co-profiling of miRNA and mRNA levels in bulk cell populations^{1–4}. This kind of work is relevant to this study as it uses a different source of miRNA and mRNA variability which could be used for some of the analysis done by the authors on their single cell data.

Response: We have now cited the references as the reviewer suggested. We acknowledge the previous efforts to understand relationships between miRNAs and mRNAs in bulk populations, but want to emphasize that this is the first study to examine such relationships in single cells that relates to the intrinsic intercellular heterogeneity of gene expression.

Comment 2: The number of single cells used for the various experiments is very low when taking into account the throughput of modern single cells analysis tools. Specifically, I would expect an order of magnitude more single cells used for the majority of the co-profiling analysis while the authors use only 19 cells. Furthermore, when assessing the reproducibility of their methods, the authors measure the correlation of miRNA and mRNA expression profiles from only a single cell. It would be more convincing to present such analysis performed on multiple single cells.

Response: The reviewer is concerned how reproducible the half-cell materials represent single cell gene expression, given that the reproducibility of miRNA and mRNA expression profiles are only from a single cell. We thank the reviewer for this comment. To address the reviewer's concern, we provide the following new data in the revised manuscript that support overall even and reproducible splitting of materials from single cells.

- (1) We performed additional small RNA library preparation and detection using half-cell materials from the same single cell. We show data from 5 single cells, covering three different cell types (human K562 and 293T cells and mouse BaF3 cell). The R^2 range from 0.8829 to 0.9394, with an average of 0.9161. These data are shown in **Supplementary Figure 2**.

- (2) We also performed half-cell splitting of lysate followed by qRT-PCR detection. A total of 30 K562 cells were subjected to this experiment. We measured the expression of miR-146b-5p and show that the technical variability (including the variability of half-cell lysate

splitting and the variability of qRT-PCR) is smaller than that of biological variability we observed (**Supplementary Figure 3**).

The reviewer also commented that the number of single cells analyzed in our study (19 cells) is fewer than many other recent single cell studies. We want to point out that the purpose of our study is very different from the purpose of most other single cell studies—most of those studies were designed to identify or classify cell populations in complex tissues with a heterogeneous mixture of cell types, and having a small number of cells is insufficient to achieve their goals. In contrast, our purpose is to study molecular relationships between miRNA and mRNA in single cells, and we do not want to have cell type heterogeneity to confound the interpretation. That’s why we chose a cell line. To address in part this question, we already showed at least in this study 19 half-cell transcriptomes quantitatively resembles the different transcriptional states measured from 400 single K562 cells with a high-throughput method (Figures 3c&d). We further demonstrate that even data from 19 single cells is sufficient to reveal regulatory relationships (Figures 4&5). We did acknowledge and discuss that our current approach is not yet a high-sample-throughput approach and future development can increase the throughput.

Comment 3: Given the previously established methods for mRNA and miRNA sequencing used by the authors, they do not present any major technological advancement beyond the optimization of the usage of half-cells.

Response: The technical novelty in our work is on the use of half-cell materials to achieve co-profiling to overcome incompatible protocols for mRNA and miRNAs sequencing. We did not claim that our technical procedure for either mRNA or miRNA sequencing is superior to other methods, but we do show a path toward single-cell miRNA/mRNA co-profiling with half-cell materials, which have not been examined by other methods and unanticipatedly not a straightforward extension of previous methods either. As shown in Supplementary Figure 1, we

need to modify the standard protocol and further optimize the workflow to ensure half-cell data faithfully recapitulate single-cell miRNA or mRNA expression. Additionally, paired-half-cell data from our method can be used to interrogate the biological relationship between miRNAs and mRNAs, which has never been shown before.

Comment 4: The authors should not conclude that the half cells from the same cell are more similar than the half cells from other cells when they have only one cell used for the half cells from the same cell. Multiple pairs and a proper statistical analysis should be applied. Furthermore, in order to make the claim of half-cells from the same cell being more similar than half cells from different cells, as shown in Fig. S4, the authors need multiple pairs of half-cells from the same single cells.

Response: We thank the reviewer for this comment. We have performed new experiments on additional half-cell pairs, as detailed in response to Reviewer 1's comment 2, and demonstrated the R^2 values (**Supplementary Figure 2c**). We also show that the R^2 between different single cells are lower than the R^2 between half-cells in three cell types, arguing for higher variability between single cells (**Supplementary Figure 4**). We further demonstrate, using single cell q-RT-PCR (**Supplementary Figure 3**) that the variability across half cells from the same single cell is less than the variability across different single cells. Nevertheless, we acknowledge the critique by the reviewer and accordingly modified our statement to "suggest the existence of intercellular heterogeneity".

Comment 5: I would expect the authors to compare their single cell measurements for miRNA and mRNA levels to bulk measurements performed on the same cell line in the same conditions.

Response: Single cell data, due to increased amplification, are not always directly comparable to bulk RNA data. To confirm the variability of miRNA and mRNA gene expression from our single cell approach, we have chosen to validate using qRT-PCR (**Figure 2C**) or another single cell RNAseq approach (**Figure 3**).

We did not perform the bulk experiment at the same time of the half-cell experiment. Nevertheless, we did compare to bulk K562 data obtained previously in our lab at a different day of culture, and profiled with the traditional Illumina small RNA sequencing protocol on bulk RNA samples. The data are shown below comment 6 for the reviewer.

Comment 6: The authors mention the dataset from Vaz et al. 20105 when comparing the expression levels of miR-92a. It would be appropriate to show how their mean miRNA expression levels, computed from the single cell measurements, compare to the complete dataset. A scatter plot and a correlation analysis should be presented. This should reveal how well the author's miRNA measurements agree with external data.

Response: It is well known that the K562 cells from different sources are different in both genetics and gene expression, because this cell line has been independently maintained throughout the research community in many labs. In addition, it is well appreciated that for small RNA sequencing, different batches of experiments with the same protocol are not always

directly comparable, not to mention different protocols. So we do not expect to see a good correlation with such published datasets with our single cell data.

We did compare to bulk K562 data obtained previously in our lab at a different day of culture, and profiled with the traditional Illumina small RNA sequencing protocol on bulk RNA samples. Data are shown below, in which the miRNA with highest expression levels is miR-92a and consistent between half-cell and bulk data. It is well appreciated that the difference in culture date and difference in library-preparation protocol will introduce variation in profiles. With this said, we did observe a trend of correlation between the bulk and half-cell data.

Comment 7: When validating the variability in miRNA expression using qRT-PCR (Fig. 2c), a direct comparison between the qRT-PCR and the half-cell miRNA sequencing results is lacking. Furthermore, I would expect the qRT-PCR to be performed on half cells as well, for a valid comparison. Moreover, a comparison of the expression means should also be performed.

Response: Following reviewer's suggestion, we performed half-cell qRT-PCR analysis to show that half cells from the same single cells had lower variability than biological variability for different single cells on miR-146b-5p expression (**Supplementary Figure 3**).

The reviewer also asked for a direct comparison of the means between qRT-PCR and small RNA sequencing on the single cell level. We show in **Figure 2c** and **Figure 2b** that for four miRNAs (miR-146b, miR-92a, let-7a, let-7i), the trend is overall the same between qRT-PCR and small RNA sequencing data: higher expressed miRNAs detected in sequencing also tend to be detected higher with qRT-PCR. We want to point out that we do not expect the means of these data to completely match up, because experimentally, neither qRT-PCR nor small RNA sequencing allows the perfect preservation of ratios between different miRNAs. Rather, the relatively levels of the same miRNA detected between samples are trustworthy.

Comment 8: It is not clear to me how the analysis performed in Fig. 2d shows that the variations in miRNA expression could not be fully attributed to genetic differences but instead associated with non-genetic cellular heterogeneity. The interpretation of the results should be clarified in the text.

Response: Following the reviewer’s suggestion, we have added clarification to the revised text. First, we intentionally used the cell lines such as K562 that are generally considered as genetically identical. Second, to “isolate” potential genetic alterations, we clonally expanded single cells because clonal expansion of a single cell will preserve all genetic variations in the initial single cell into all cells within the clone, and because clonal expansion of a single cell over several divisions is unlikely associated with major genetic alterations during the expansion process. Comparing Figure 2d to 2c, the observed “suppression” of cell-cell variability after clonal expansion cannot be attributed to the genetic variation existed in the initial single cells before clonal expansion.

Comment 9: The quality check for the RNA-seq data from 400 cells performed by the authors should also include a direct comparison of gene expression levels and variability with the data generated by InDrop.

Response: Following the reviewer’s suggestion, we include the analysis of gene expression level and variability from the 400 K562 cell profiles (bottom panel of **Supplementary Figure 8**). Of note, 6167 genes were detected in this 400 cell dataset (based on the Dropseq-like approach) as compared to 13788 genes detected in the 19 half-cell RNAseq dataset.

Comment 10: It is not clear to me how the analysis in Fig. S10 was performed and how it shows a correlation between transcriptional clusters and top ranked miRNAs. On what values was the clustering performed? Why were specific clusters attributed to the annotated miRNAs? How does the heatmap convey the desired message?

Response: We agree the reviewer that the old Fig S10 does not convey a key message and is not essential for our conclusions. So we have removed that Figure in the revised manuscript to avoid distraction.

Comment 11: "indicating the possibility for the top-ranked miRNAs to not only regulate specific targets but also influence global transcriptional phenotype in single cells." . The authors imply causation where only a correlation is observed.

Response: We thank the reviewer for pointing this out. Indeed, we do not have data to firmly conclude on causality. We have modified the sentence to use “suggesting” instead of “indicating”.

Comment 12: "For let-7a-5p and let-7i-5p, BCL2L1 decreased as let-7 expression increased at the high end of miRNA expression, suggesting a possible role for let- 7a/let-7i to alter the level of its target mRNA BCL2L1. Such correlation was not seen at the low end of let-7a/let-7i expression levels, as expected." For let-7a-5p one can also observe that BCL2L1 increased as let-7 expression increased at the low end of miRNA expression. The authors seem to report their interpretation of the data in a biased manner.

Response: The reviewer is referring to data in the old Figure 4f. We agree with the reviewer that such single gene examples could be biased and not as strong as the genome-level analysis. The genome-level analyses are shown in Figure 4a-4e, which already delivered the message. As such, we have removed the old Figure 4f and associated text from the revised manuscript.

Comment 13: The author's results support the notion that a negative relationship between miRNAs and their targets can be observed in their half cell data. However, in order for them to claim a functional relationship, as they do, it is not sufficient to associate these relationships with clustered transcriptional programs, in my opinion. A functional relationship could be shown by experimental association of observed relationships with some measurable phenotype such as apoptosis markers, cell cycle phase or differentiation markers.

Response: We thank the reviewer for appreciating our result that a negative relationship between miRNAs and their targets can be observed in our half-cell data, which is one of our points. We want to clarify that we do not intend to claim that intercellular miRNA variability determines cellular functions (although we cannot exclude that possibility either). We have modified the text to avoid misunderstanding on this front.

Comment 14: The authors use their data to explore regulatory relationships controlling miRNA expression. It is true that this exploratory analysis triggered the identification of a regulatory relationship for miR-146-5p, followed by verification on bulk cultures as opposed to single cells. However, comparative analysis of bulk miRNA and mRNA expression data across cell lines or a single cell line in different conditions could be used as an alternative starting point. The use of cell to cell variability in miRNA expression levels could, in my opinion, be replaced by any other source of miRNA expression variability. Therefore, I fail to note a significant advantage of the author's methods in uncovering such regulatory relationships.

Response: We agree with the reviewer that we cannot exclude the possibility that the regulatory relationship between AKT and miR-146-5p could be obtained by other techniques, such as by studying gene expression variability in bulk samples.

We want to point out that the studies based on bulk samples will not allow us to understand and directly examine what controls intercellular miRNA variability. To investigate this, we performed a new experiment in which single-cell miR-146b-5p expression levels were determined after cells were treated with vehicle control or AKT inhibitor. Data show that the variation in miR-146b-5p (reflected by standard deviation) was reduced after AKT inhibition (**Supplementary Figure 11b**), suggesting that AKT activity controls the variability of this miRNA in single cells.

Comment 15: "This was realized with a halfcell genomics approach, which overcomes the challenge to separate and capture small RNAs from mRNA without introducing material loss and technical variation." A reference or supplementary data for these challenges should be included since this claim poses the main motivation for using the author's methods as opposed to potential alternatives.

Response: We have cited reference as requested.

Minor issues:

1) In Fig. S1 x and y axis labels are inconsistent.

Response: We thank the reviewer for spotting this difference. We have corrected the axis labels.

2) Given the optimization of lysis conditions presented in Fig S1, it is not clear which method was used for the generation of Fig 2a.

Response: It is using the improved method, which we clarified in the figure legend.

3) "We further examined the validity of this half-cell approach using different cell types and a different measurement techniques.". However, the authors only show one different cell type with one different measurement technique.

Response: We have modified this sentence in the revised manuscript. Since we have now more cell types, we also changed wording regarding the measurement technique.

4) In Fig. S5 and S6 the usage of a divergent colormap is not justified. A sequential colormap should be used. Moreover, the usage of a pure blue color for self-correlations while the lower correlations use blue hues is a very poor choice.

Response: Following reviewer's suggestions, we have changed the color maps in **Supplementary Figure 6 and 7** (old Fig S5 and S6), so that they are on the same color scale. Additionally, we have used grey to indicate self-correlation.

5) In Fig. S5 there is one cell that seems to have higher correlations with all other cells, how can this be explained?

Response: It seems that the cell the reviewer indicated (the last cell) had good correlation with other cells in both miRNA and mRNA data (**Supplementary Figure 6 and 7**). Not the order of the cells is the same in both figures. Although we do not know exactly why, we speculate that this particular cell may share transcriptional features across multiple transcriptomic clusters.

6) "Comparing intercellular variability across miRNAs, both miR-146b-5p and let-7i-5p showed higher variability than miRNAs with similar expression levels." The authors should accompany these claims with statistical analysis.

Response: we have indicated the standard deviation in the text.

7) MiR-let-7a is missing from Fig. 2d.

Response: We did not measure let-7a in Figure 2d, because Figure 2d was meant to examine miRNAs with high intercellular variability. Let-7a has a relatively low variability in Fig 2c. That is why we did not perform detection on let-7a in Figure 2d.

8) In Fig. 4f the x-axis should also be labeled and not just explained in the legend.

Response: We have removed Figure 4f, as explained in response to Comment 12.

9) In the context of Fig. 4f, the correlation coefficient and the p-value for the correlation of miR-125a-5p with BCL2L1 should be reported.

Response: We have removed Figure 4f, as explained in response to Comment 12.

Reviewer #2

Comment 1: The manuscript is a follow-up on the authors' previous paper from 2017 (ref. 2 of this manuscript), in which the authors developed a small RNA sequencing method for single cells. Unlike established methods, where genomic DNA and mRNA are co-analyzed, they present an innovative technology for parallel sequencing of both mRNA and small RNA of the same single cell, in order to gain new insights into post-transcriptional modulation of non-genetic intercellular heterogeneity by miRNAs and how miRNA variations might be affected by protein-coding genes. To this end, the authors split cell lysates into halves and perform the two incompatible sequencing procedures (their own small RNA sequencing technology and the commercial SMART-Seq protocol for mRNAs) separately on these halves. They first validate this approach and then apply it to identify a yet unknown regulation of miR-146b-5p by the AKT pathway.

Major criticisms and concerns

1. Novelty, methods and foundation of claims.

Parallel analysis of miRNA and mRNA has not been done in the same single cells has not been published so far. In this regard the approach presented in the paper is novel. However, the method is entirely based on the half-cell approach and I am missing data on the reproducibility of the splitting method, as the whole technology rests on the assumption that miRNAs and mRNAs are homogeneously distributed between both halves of a single cell.

a. The authors are showing the correlation between two halves of the same cell only for three single cells in the paper (in Figure 2a and Supplementary Figures 1 and 2). I am certain they tested this on more than three cells. I would like to know how many times these experiments were replicated and to see the average correlation of halves from the same cell as proof that the approach is valid and that the high correlation they observed was not random.

b. Heterogeneity may have several sources. For a general reproduction of the data by the readership, important information on the methods are missing. For example, what kind of PCR tubes and pipette tips were the authors using for their experiments? This is not mentioned. I think this is relevant here, since nucleic acids tend to stick to normal, uncoated plastic. When working with bulk RNA from a large number of cells this does not matter, of course, but it is a major issue when working with single cell material, because it may cause significant loss of said material.

c. Another way to address the origin of heterogeneity (true cell-to-cell heterogeneity or technical noise) is the use of spike-in experiments. This could be done for instance by conducting a series of spike-in experiments with artificial miRNA molecules introduced at single-cell levels in defined amounts. Provided that half-cell sampling approach is unbiased and miRNA amplification efficient across the physiological range of miRNA expression the expected outcome would be: (i) equal amount of spike-ins detected in both halves of the single-cell lysate (applies to spike-in representing all – high, medium and low – miRNA expression levels) and (ii) stable and reproducible quantities or reads corresponding to each spiked miRNA, irrespectively of the initial amount of the molecules introduced in the lysate.

Response: We thank the reviewer for noting the novelty of our study.

a. The reviewer is concerned how reproducible the half-cell materials represent single cell gene expression, given that the reproducibility of miRNA and mRNA expression profiles are

only from a single cell. We thank the reviewer for this comment. To address the reviewer's concern, we provide the following new data in the revised manuscript that support overall even and reproducible splitting of materials from single cells.

- (1) We performed additional small RNA library preparation and detection using half-cell materials from the same single cell. We show data from 5 single cells, covering three different cell types (human K562 and 293T cells and mouse BaF3 cell). The R^2 range from 0.8829 to 0.9394, with an average of 0.9161. These data are shown in **Supplementary Figure 2**.

- (2) We also performed half-cell splitting of lysate followed by qRT-PCR detection. A total of 30 K562 cells were subjected to this experiment. We measured the expression of miR-146b-5p and show that the technical variability (including the variability of half-cell lysate splitting and the variability of qRT-PCR) is smaller than that of biological variability (**Supplementary Figure 3**).

- b. According to the reviewer's suggestion, more relevant details have been added in Materials and Methods section, such as information on tubes etc. Also included are catalog numbers.
- c. The reviewer is concerned whether the variability in miRNA expression is due to true biological intercellular heterogeneity or due to technical variability, and suggested the use of spike-in controls. Our results with additional (see point **a** above) pairs of half-cell miRNA profiles further support relatively low technical variability in miRNA profiles. Our data on qRT-PCR validations of intercellular miRNA variability (**Figure 2c**) solidifies our findings of intercellular variability of miR-146b and let-7i in K562 cells.

To further solidify our results that there exists intercellular variability associated with miRNA/mRNA expression, we show new data (**Supplementary Figure 11b**) that inhibiting AKT reduces the intercellular variation of miR-146b, further confirming the true variable nature of miR-146b in single K562 cells and such variability can be modulated.

Regarding the suggestion of using spike-in controls, we agree that this is an interesting thought that we should attempt in the future. However, the single cell field has debated whether spike-in controls are truly faithful measures of technical variability, as spike-in controls tend to be “naked” RNA, whereas cellular RNAs may be bound by proteins.

Comment 2: Methodological and functional validation

In general, I found the functional validation experiments elegant and interesting. However, the paper would definitely benefit from further validation, i.e. by testing the detected relationships in independent model system (e.g. multiple cells lines) or primary cells. Basically, the authors have used only one cell line (K562) and validation of the general approach with at least additional cell lines (if not clinical samples for the general aspects, such as feasibility with “real life” miRNA and mRNA levels) would be important.

Response: We have followed the reviewer’s suggestion, and performed additional experiments.

Specifically, (1) we have validated that in MCF-7 cells, which also express miR-146b, AKT inhibition leads to a similar increase in miR-146b-5p expression (**Supplementary Figure 11a**). (2) we also performed new experiments to show that other AKT inhibitors produced similar results, to avoid specificity concerns of a single AKT inhibitor (**Supplementary Figure 11c, 11d**).

Comment 3: Graphical representation of data.

It is repeatedly not clear what type of data was presented in the figures. For clear and complete understanding of the paper comprehensive explanation of the data presented in the figures is essential. For instance, a term “normalized and log₂-transformed miRNA expression levels” does not explain precisely what kind of data was depicted in the plot.

Response: We apologize for the problems in figure legends. We have added details into figure legends. Where appropriate, we refer to methods section for additional details, including the normalization and log₂-transformation of miRNA expression.

Comment 4. Sequencing performance of the new workflow.

The manuscript does not state how the success of single-cell miRNA sequencing was defined. No quality metrics were provided addressing the performance of either single-cell miRNA or mRNA sequencing – e.g. the amount/proportion of reads mapping to other portions of the genome (different to expected miRNA and mRNA sequences), amount/proportion of non-mappable reads, etc.

Response: Following the reviewer’s suggestion, we have added details of mapping to other RNA species in the genome to the revised manuscript for miRNAs, in the Methods Section. We also detail the mapping strategy in the Methods Section. Mapping performance of RNAseq data, which uses SmartSeq v4, was similar to other single cell RNAseq runs using this kit. We have indicated the number of detected genes from each of the library in the Methods section.

Minor Comments:

There are numerous syntax errors or missing words throughout the manuscript. Many of these errors reduce legibility and should be corrected.

Response: We thank the reviewer for catching these issues. We have read through and corrected them.

1. The y-axes of both plots of Supplementary Figure 1 have typing errors in their labels: “loh2” instead of “log2”.

Response: We thank the reviewer for catching this mistake. We have corrected the error.

2. There is a typing error in the x-axis label of Supplementary Figure 2.

Response: We thank the reviewer for catching this mistake. We have corrected the error.

3. Some graphs are in not precisely labeled, e.g. “Log2 (miRNA Level)”. What is the miRNA level exactly? In some graphs the labels actually say “FPKM”, but often the units are missing.

Response: We have detailed such units in the figure legends, including reference to Methods. The miRNA level refers to normalized fraction of miRNA within total miRNAs. This is now clarified in the Methods Section, with reference to this section in the figure legends.

4. Supplementary Figure 7: I do not consider library size and number of expressed genes suitable measures of sequencing quality. Please, add average PHRED score distribution across reads.

Response: We apologize for the confusion. **Supplementary Figure 8** (old supplementary figure 7) is meant to demonstrate the quality of the library (i.e. how many genes are detected and how many transcripts are detected), not the quality of the sequencing reads. We have modified the figure legends to avoid possible misinterpretation.

5. Supplementary Figure 8 needs to be improved for better legibility. The genes need to be sorted in the same way and need to have the same colors in both panels. Right now, it takes too much time to find the same gene in both panels, in order to compare its expression.

Response: We have replotted **Supplementary Figure 9** (old supplementary Figure 8) to be in the same order. The old figure contained the top 20 most highly variable genes from each dataset which are not fully overlapping. So in this revised figure, we have included the 11 common genes within the top 20 most highly variable genes from the two datasets.

6. Figure 4f should have a label for the x-axes. The explanation in the legends is not enough.

Response: We have removed Figure 4f, because it is not essential for our conclusions and may add confusion. The details are explained in response to Reviewer 1 Comment 12.

Reviewer #3

Overall Comment: The manuscript by N. Wang and colleagues refers to co-sequencing of both mRNAs and microRNAs from the same single cell by implementing a half-cell approach. The authors developed a strategy to co-profile miRNAs and mRNAs from a single-cell lysate split into two half-cell fractions. The lysates then underwent either miRNA or mRNA transcriptome sequencing.

The manuscript is overall well written. As far as I know, this is the first attempt to combine genome-wide profiling of both mRNA and post-transcriptional regulators from the same single cell and I think the study will be of great interest to the community.

Response: We thank the reviewer for considering our study to be of great interest to the community.

Main Comment: My major claims, however, involve the comparison between the two half-cells, both sequenced for miRNAs or mRNAs, and the whole miRNA or mRNA single-cell sequencing.

The entire work is based on the assumption that the half-cell sequencing procedure is working properly based on only one single cell experiment (figure 2a and 3a). Since sampling molecules from a fixed volume and sampling from half of this volume is in principle very different, I think that in order to show that one half-cell is really representative of the other half, the authors should show results for more than one single cell with both halves sequenced for miRNAs or mRNAs. Said differently, they should show that the half-cell sequencing results presented in Figures 2a and 3a are reproducible in terms of correlation (and PCA analysis for miRNAs (suppl. Fig.4)) between the two halves for more than one cell (that is, for a number of cells comparable to the 19 presented) for both mRNA and miRNAs.

Response: The reviewer is concerned how reproducible the half-cell materials represent single cell gene expression, given that the reproducibility of miRNA and mRNA expression profiles are only from a single cell. We thank the reviewer for this comment. To address the reviewer's concern, we provide the following new data in the revised manuscript that support overall even and reproducible splitting of materials from single cells.

- (1) We performed additional small RNA library preparation and detection using half-cell materials from the same single cell. We show data from **5** single cells, covering three different cell types (human K562 and 293T cells and mouse BaF3 cell). The R^2 range from 0.8829 to 0.9394, with an average of 0.9161. These data are shown in **Supplementary Figure 2**.

(2) We also performed half-cell splitting of lysate followed by qRT-PCR detection. A total of **30** K562 cells were subjected to this experiment. We measured the expression of miR-146b-5p and show that the technical variability (including the variability of half-cell lysate splitting and the variability of qRT-PCR) is smaller than that of biological variability (**Supplementary Figure 3**).

In addition to the experiments above, we also performed a new experiment to examine the impact of AKT inhibitor treatment on the intercellular variability of miR-146b-5p expression. miR-146b-5p is a variable miRNA in K562 cells as determined by our half-cell small RNA sequencing. The data in **Supplementary Figure 11b** not only confirmed the existence of

intercellular expression variation for this miRNA, but also revealed that AKT pathway inhibition leads to reduced intercellular variation of this miRNA. These data further help to confirm the validity of our half-cell miRNA profiles.

Minor Comment 1: With respect to Figure 2a-b, why is the number of annotated miRNAs different in the two panels? If so, the authors should show panel (a) with the same annotated miRNAs as panel (b) (or at least justify the difference between the two).

Response: Figure 2a and Figure 2b has the same number of miRNAs plotted. The difference is that Figure 2a is a plot between two halves of a single cell, so that there are many miRNAs with the lowest expression levels in both halves, resulting in such dots overlapping each other at the bottom left corner. In contrast, Figure 2b is a plot reflecting variation across 19 half cells. This leads to more separation of dots with low expression levels.

Minor Comment 2: Few typos: In Figures 2b “Standard deviation” is written SD while in figure 3b is “Std”; In the text the authors always refer to Pearson coefficient r while in the figures they show R^2

Response: We thank the reviewer for pointing these out. We have now corrected these typos/inconsistencies.

Reviewers' comments:

Reviewer #1 (Remarks to the Author):

The authors performed various experiments and corrections to the paper in order to satisfy most of the comments provided by the reviewers. Most notably, they addressed the issue of reproducibility of miRNA and mRNA expression profiles. The additional data presented supports the reproducibility of the half-cell method across cell lines. Moreover, the authors acknowledged the limited throughput of their method and showed data supporting the claim that their current data is sufficient to reveal regulatory relationships. However, I noticed a couple of minor issues in the authors' response to reviewers:

- In the response to comment 7 the authors write: " We want to point out that we do not expect the means of these data to completely match up, because experimentally, neither qRT-PCR nor small RNA sequencing allows the perfect preservation of ratios between different miRNAs. Rather, the relatively levels of the same miRNA detected between samples are trustworthy". The relative levels should still be compared using a spearman correlation.
- In the response to comment 14 the authors present the reduction in variation of miR-146b-5p expression upon inhibition of AKT as shown in their Supplementary Figure 11b. This result should be accompanied by a statistical analysis and the significance should be reported.

Reviewer #3 (Remarks to the Author):

I appreciate the effort of the authors to show the reproducibility of their half-cell sequencing procedure, however I think my comments were not fully addressed.

Major comments:

- i) data for the reproducibility of half-cell sequencing procedure for mRNA are still missing;
- ii) in supplementary figure 2 the authors show data from two out of five analysed cells (panels a and b) for miRNA sequencing. All the data should be shown.

Summary:

We thank the reviewers for valuable comments that helped to further improve this manuscript. All the changes in the revision (main text) are highlighted in red, and the responses to the review report are summarized below.

Point-by-point response to reviewers:

Reviewer #1

Comment: The authors performed various experiments and corrections to the paper in order to satisfy most of the comments provided by the reviewers. Most notably, they addressed the issue of reproducibility of miRNA and mRNA expression profiles. The additional data presented supports the reproducibility of the half-cell method across cell lines. Moreover, the authors acknowledged the limited throughput of their method and showed data supporting the claim that their current data is sufficient to reveal regulatory relationships. However, I noticed a couple of minor issues in the authors' response to reviewers:

Response: We thank the reviewer for appreciating the improvements in our first revision and recognizing the importance of this work. We have addressed the minor issues as the following.

- In the response to comment 7 the authors write: " We want to point out that we do not expect the means of these data to completely match up, because experimentally, neither qRT-PCR nor small RNA sequencing allows the perfect preservation of ratios between different miRNAs. Rather, the relatively levels of the same miRNA detected between samples are trustworthy". The relative levels should still be compared using a spearman correlation.

Response: In response to the suggestion of the reviewer, we have conducted both Pearson and Spearman correlation analyses. Below is the comparison of miRNA expression determined by single-cell qRT-PCR and that determined by single-cell miRNA sequencing. The means of single cells for four miRNAs, hsa-miR-92a-3p, has-miR-146b-5p, has-let-7a-5p and has-let-7i-5p are plotted. R value for Pearson correlation is 0.87; $p=0.131$. R value for Spearman correlation is 0.80; $p=0.333$.

- In the response to comment 14 the authors present the reduction in variation of miR-146b-5p expression upon inhibition of AKT as shown in their Supplementary Figure 11b. This result should be accompanied by a statistical analysis and the significance should be reported.

Response: We thank the reviewer for the suggestion. In the revised manuscript, we have included a statistic analysis and the significance is reported.

For Supplementary Figure 11b (AKT inhibitor effect on miR-146b-5p expression variation in single cells), we determined that $P=0.006$ ($P<0.01$). This is now reported in the main text and corresponding figure of the revised manuscripts. Also see below. The method used to perform the p-value calculation is included in the revised Methods under Statistical Analysis.

Reviewer #3

Comment: I appreciate the effort of the authors to show the reproducibility of their half-cell sequencing procedure, however I think my comments were not fully addressed. Major comments:

Response: We thank the reviewer for appreciating our efforts in the first revision. The two comments have been addressed.

i) data for the reproducibility of half-cell sequencing procedure for mRNA are still missing;

Response: We performed additional half-cell mRNA sequencing measurements on the same single cells in human 293T or murine NIH3T3 cells, the statistical analysis was conducted and the significance like R^2 was calculated. Data are shown below and also included in the revised Supplementary Figure 2b.

ii) in supplementary figure 2 the authors show data from two out of five analysed cells (panels a and b) for miRNA sequencing. All the data should be shown.

Response: Following the suggestion of the reviewer, we now provide all half-cell miRNA data from the same single cells in the revised Supplementary Figure 2a (also see below). Please note that one of the cells are shown in main Figure 2a, and four additional cells are shown in Supplementary Figure 2a, with a total of 5 single cells.

We believe all the questions have been fully addressed and again are grateful to the reviewers for their insightful comments. Looking forward to your editorial decision soon.

REVIEWERS' COMMENTS:

Reviewer #3 (Remarks to the Author):

The authors fully addressed my previous major remarks and I think the manuscript is now suitable for publication.